# Forearc density structure of the overriding plate in the northern area of the giant 1960 Valdivia earthquake.

Andrei Maksymowicz[1], Daniela Montecinos-Cuadros[1], Daniel Díaz[1], María José Segovia[1], Tomás Reyes[2,3]

[1]Departamento de Geofísica, Universidad de Chile, Blanco Encalada 2002,  Santiago, Chile
[2]Departamento de Geología, Universidad de Chile, Santiago, Plaza Ercilla 803, Chile
[3]Instituto de Geocronología y Geología Isotopica (INGEIS-CONICET), Universidad de Buenos Aires (UBA), Argentina.

*Correspondence to*: Andrei Maksymowicz (andrei.maksymowicz@uchile.cl)

**Abstract.** The objective of this work is to analyse the density structure of the continental forearc in the northern segment of the 1960 Mw 9.6 Valdivia earthquake. Regional 2D and local 3D density models have been obtained from available gravity data in the area, complemented with new gravimetric stations. Models are constrained by independent geophysical/geological information and new TEM and MT soundings. The results show a segmentation of the continental wedge along and perpendicular to the margin, highlighting a high-density anomaly, below the onshore forearc basin, that limits the Late Paleozoic-Early Mesozoic metamorphic basement in the region where Chaitenia terrain has been proposed. A progressive landward shift of this anomaly correlates with the high slip patch of the giant 1960 Mw9.6 Valdivia earthquake. Based on these results, we propose that the horizontal extension of the less rigid basement units conforming the marine wedge and Coastal Cordillera domain could modify the process of stress loading during the interseismic periods, and also that changes in position and extension of the Late Paleozoic-Early Mesozoic accretionary complex could be linked with the frictional properties of the interplate boundary. This analysis provides new evidence of the role of the overriding plate structure on the seismotectonic process in subduction zones.

## 1 Introduction

The physical structure of the oceanic and continental plates have had an important role in the long and short-term deformation process of the subduction margins. On the other hand, the tectonic activity has modified the internal structure and geometry of the tectonic plates (i.e., Bilek et al, 2003; Hackney et al., 2006; Hicks et al., 2014; Contreras-Reyes and Carrizo, 2011; Bassett and Watts, 2015; Poli et al.,2017). This geodynamical feedback is evinced by spatial correlations between the physical segmentation of the continental wedge, and ruptures of large megathrust earthquakes (i.e., Contreras-Reyes et al. 2010; Li and Liu, 2017; Martínez-Loriente et al., 2019; Molina et al., 2021). Examples of this is the spatial correlation between gravity (density) anomalies in the continental wedge and the location of high slip patches in large earthquakes (>7.5-8 Mw, Song and Simons, 2003, Wells et al., 2003; Álvarez et al., 2014; Bassett and Watts, 2015; Bassett et al., 2016; Schurr et al., 2020), which suggests that changes in normal stresses on the seismogenic zone have a role on the seismic rate and slip propagation during large earthquakes (Tassara, 2010; Maksymowicz et al., 2015; 2018; Molina et al., 2021). On the other hand, changes of the continental wedge geometry have been associated to variations of the interplate boundary friction at the maximum slip patches of the large 2011 Mw9.0 Tohoku-Oki, 2010 Mw8.8 Maule and 1960 Mw 9.6 Valdivia earthquakes (Cubas et al., 2013a,b; Maksymowicz, 2015; Contreras-Reyes et al., 2017; Molina et al., 2021).

Diverse works have highlighted the importance of the transition between accretionary prism (or highly fractured frontal units) and the more rigid rocks of continental basement as a tectonic limit, controlling, at least partially, the upward propagation of coseismic slip, foreshocks and aftershocks during large megathrust earthquakes (Scholz, 1998; Contreras-Reyes et al., 2010;

Moscoso et al., 2011; Kodaira et al., 2012; León-Ríos et al., 2016; Maksymowicz et al., 2017; 2018; Tsuji et al. 2017). At the same time, the downdip limit of the megathrust earthquakes has been related (among other factors) to physical properties of the mantle wedge and deep interplate boundary (Peacock and Hyndman, 1999; Seno, 2005; Wang et al., 2020), which are modified by fluids subduction, slab dehydration and presence of basal accretionary complexes (Moreno et al., 2018; Menant et al., 2019). Notwithstanding, less attention has been paid to the internal physical structure (and lithology) of the continental crust above the downward limit of the megathrust, even considering that all forearc units above the fragile/ductile limit should work as a part of the same mechanically coupled system (van Dinther et al., 2012; Comte et al., 2019).

In this context, we have explored the continental forearc density structure of the Nazca-South America subduction zone in a segment where the high slip patch of the giant 1960 Mw9.6 Valdivia earthquake ruptured (Fig. 1). As aforementioned, this slip patch correlates not only with a low gravity anomaly above the marine forearc (Wells et al., 2003) and low continental slope angles (Maksymowicz, 2015), but also, with a landward extension of Palaeozoic metamorphic outcrops onshore (Fig 1a). Furthermore, ages and petrological data of continental basement rocks (metamorphic and plutonic rocks) suggest a complex ancient history of accreted terrains (Ramos et al., 1986; Rapalini, 2005) that constitutes the presents continental crust in the area (Fig. 1b). Particularly, the recent proposal of an oceanic terrain accreted against Gondwana margin at Devonian time (Chaitenia, Hervé et al., 2016; 2018, Ct in Fig. 1b) could determine changes in the internal structure of the continental forearc, southward of ~40ºS. However, the exact limits of this basement configuration remain poorly constrained. In order to reveal the crustal structure of this active portion of the Chilean margin, this work presents the results and interpretation obtained from regional 2D density models, extended from Nazca plate to the Andes Cordillera (Fig. 3), and a local 3D density inversion of the continental forearc (red rectangle in Fig. 2, and 3). The models include magnetotellurics (MT) and transient electromagnetic (TEM) measurements, as well as available independent geophysical and geological data to constraint forward modelling and 3D inversion.

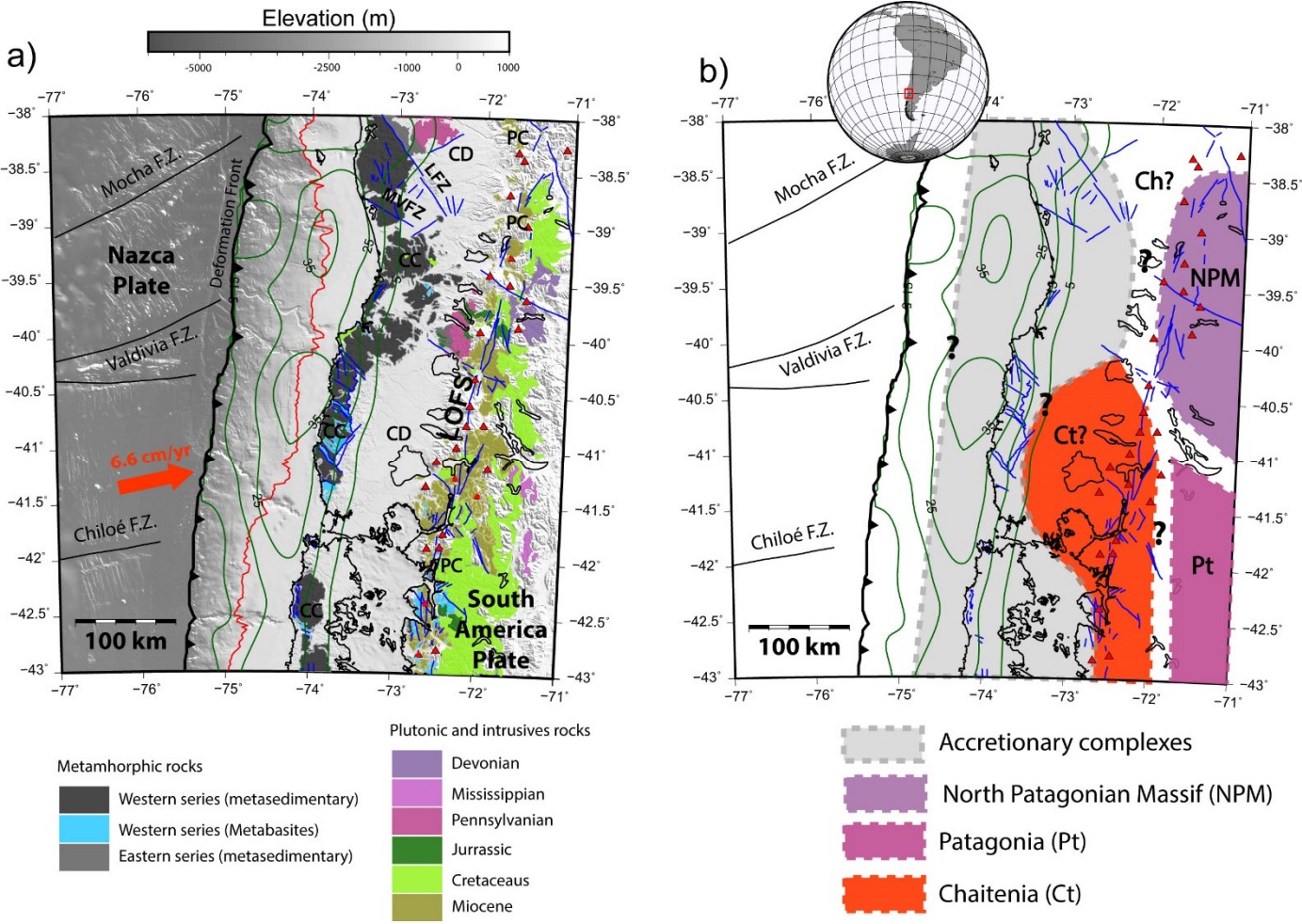

**Figure 1: Geotectonic settings of the studied zone. a) Tectonic and morphostructural features above the coulored Bathymetric/topographic elevation grid. Metamorphic and plutonic/intrusives outcrops are shown. Oceanic fracture zones (Mocha, Valdivia and Chiloé) are indicated in black. Deformation front (continental wedge toe) at the Nazca-South America trench is indicated by a bold black line. The blue lines correspond to continental structures identified at the surface (SERNAGEOMIN, 2003; Melnick and Echtler, 2006), including: Liquiñe-Ofqui fault system (LOFS), Mocha-Villarrica fault zone (MVFZ) and Lanalhue fault zone (LFZ). The red triangles illustrate active volcanoes. The red arrow indicates the direction of Nazca-south America convergence and the green lines represent the iso-slip contours of Valdivia earthquake according to Moreno et al, 2009. b) Schematic map of basement units after Hervé et al., (2018) and other elements as in a).**

## 2 Geotectonic Settings

The study zone, located between 38.5°S and 42.5°S (Fig. 1), is part of the South-Central Chile margin, where the oceanic Nazca plate subducts beneath the continental South American plate. The current rapid convergence rate (~6.6 cm/yr, Kendrik et al., 2003; Vigny et al., 2009) determines high seismotectonic activity, including the occurrence of mega-earthquakes as the giant 1960 Mw9.6 Valdivia earthquake (the largest instrumentally register worldwide). In the long-term, this subduction process has been continuously active since Jurassic times (Charrier et al. 2007), being superimposed to ancient tectonic processes of Gondwana structuration (Ramos et al, 1986), and generating the current configuration of the continental South America plate western border. Marine seismic studies, to the north and south of the study area, indicate that the structure of the continental wedge shows a physical and tectonic segmentation from the trench to the coast, characterized by active accretionary prims along the lower slope regions, compressional geometries and the development of confined slope basins inside the middle and upper slope, while the shelf region exhibits forearc basins with a complex deformation style structured by normal and inverted faults (Bangs and Cande, 1997; Geersen et al., 2011; Becerra et al. 2013; Bangs et al., 2020). Consistently, Vp models derived from

wide-angle seismic refraction, at 38ºS (Contreras-Reyes et al. 2008) and south of 43ºS (Contreras-Reyes et al., 2010), present changes in the deep structure of the continental wedge that can be interpreted as transitions between accretionary prism, paleo-accretionary rocks and continental basement. Regarding the geometry of the marine forearc, the continental wedge shows a narrow continental slope (defined between the deformation front and shelf break, see Fig. 1) to the south of ~41ºS. This morphological change corresponds to a decrease in the slope angle at the northern region of the Valdivia earthquake rupture (38.5ºS-41ºS), which in turn, can be interpreted as a decrease in the effective friction coefficient (μb*) at the interplate boundary (Dahlen, 1984; Cubas et al., 2013b; Maksymowicz, 2015).

Onshore, three major trench-parallel morpho-structural units from west to east can be observed (Fig. 1a): (1) The Coastal Cordillera (CC) where old rocks of a paired metamorphic belt are exposed (Hervé, 1988), (2) the Central Depression (CD) characterized by the presence of unconsolidated Quaternary sediments overlaying Cenozic deposits (Jordan et al., 2001), and (3) the Principal Cordillera (PC), where the active volcanic arc is currently located. In a close spatial relation with the volcanic arc, the prominent Liquiñe-Ofqui Fault System (LOFS, Fig. 1) stretches along more than 1.000 km long between 37°S and 46°S (Cembrano et al., 1996). This continental structure has been interpreted as right-lateral strike-slip system that currently concentrates most of the crustal intraplate seismic activity in response to oblique Nazca South America convergence (Lange et al., 2008, Orts et al., 2012) and exhumation at these latitudes (Adriasola et al., 2005; Glodny et al., 2008). Moreover, numerous tectonic lineaments and faults zones have been described (SERNAGEOMIN, 2003; Melnick and Echtler, 2006), generally showing Northwest and Northeast orientations. According to Melnick et al. (2009), the kinematics of LOFS generates intense deformation in its northern limit, explaining the deformation associated to large north-west strike continental faults (as LFZ) and the eastward bending of the CC.

Accretionary metamorphic complexes, associated to late Paleozoic-early Mesozoic subduction, are exhumed along the study zone (Hervé, 1988; Duhart et al., 2001; Willner et al., 2004; Hervé et al., 2013). These units correspond to a paired metamorphic belt, which includes the Western and Eastern Series (WS/ES) formed under high P/T and low P/T conditions, respectively. WS has been interpreted as a basal accretionary complex while ES is interpreted as a frontal accretionary prism and/or as the shallow sedimentary units deformed by the basal underplating of WS units (Glodny et al., 2005, Willner et al., 2005). This paired metamorphic belt is observed continuously at the CC, but the width of their outcrops varies along the margin (see Fig. 1a). Between ~38°S and 40ºS, and southward of ~ 41.5ºS, outcrops of WS are observed eastward, near the western limit of PC. Thus, between ~40ºS to ~ 41.5ºS, the eastern limit of these units is not defined due to the presence of the CD deposits and could form most of the forearc basement or it could be confined near the coast. Westward of accretionary metamorphic complexes and north of 38ºS, the Coastal Batholith (Late Palaeozoic intrusive rocks) is observed along CC, but southward (in the study zone) the outcrops of this ancient volcanic arc bends to the Southeast and becomes part of the PC. Younger Plutonic and intrusive rocks, related to magmatic arcs from Mesozoic to Cenozoic times (Andean tectonic Cycle), are observed along the PC near the position of the active volcanic arc and the LOFS, forming the North Patagonian Batholith (Charrier et al., 2007; Hervé et al., 2018; SERNAGEOMIN, 2003; SEGMAR, 1997, see Fig. 1a).

The continental crust of the western border of South America was configured, during Paleozoic times, by collisions of allochthonous terranes against Gondwana (Rapalini, 2005). To the north of the study zone, Chilenia terrane (Ch in Fig. 1b) collided during Devonian times (Ramos et al, 1986; Hyppolito et al., 2014, and references therein), but its southern extension is roughly defined and could be present in the northern area of the study zone. Southward, the geodynamic evolution of the margin

during Devonian to Triassic times, has been explained with a double subduction system (Hervé et al., 2016). These authors proposed the development of an island arc (named as Chaitenia, Ct in Fig. 1b) parallel to the margin colliding with Gondwana during Carboniferous times (Hervé et al., 2016, 2018, Rapela et al., 2021, Ct in Fig. 1b). If this hypothesis is correct, the continental crust of the current forearc corresponds to Chaitenia, south of ~40ºS. However, it is important to point out that the limits between all these terranes are poorly constrained in the study zone owing to the scarcity of basement outcrops.

## 3 Data and methods

### 3.1 Gravimetric database and processing

We compile a gravimetric data base (see Fig. 2), including public databases and new measurements in the studied area. The resulting merged database includes: (1) onshore gravimetric data acquired by Chilean and European institutions in the Central Andes from 1982 to 2006, originally compiled by Schmidt and Götze (2006), (2) 167 new gravimetric stations acquired by our group in 2019, (3) marine gravimetric profiles available in the GEODAS database data (NOAA) and (4) satellite gravimetric grid from Sandwell and Smith (https://topex.ucsd.edu/cgi-bin/get_data.cgi), Sandwell and Smith, 2009; Sandwell et al., 2014) to cover marine gaps and regions to the south of 42ºS. Bathymetric/topographic database merges onshore elevation grid (SRTM elevation grid, Jarvis et al., 2008) and swath bathymetry data of the studied zone (Flueh and Grevemeyer, 2005), complemented by Global Topography V18.1 (Smith and Sandwell, 1997).

The new gravimetric data were distributed to fill in some observed gaps in onshore studies, and to complement and validate gravity and topographic information from old stations. The gravity acquisition was made using a Lacoste & Romberg G-411 gravimeter with a digital upgrade (http://www.gravimeter-repair.com) funded by ANID-FONDECYT project Nº11170047. Elevation was obtained by differential GPS using Topcon HiperV instruments of the University of Chile (DGF). GPS data were processed with the permanent GPS bases of the Chilean national seismological network (Centro Sismólogico Nacional, http://www.csn.uchile.cl/red-sismologica-nacional/red-gps/) and the new gravity measurements were tied to the absolute gravity stations available in the study zone (International Gravimetric Bureau (BGI), https://bgi.obs-mip.fr/). Estimated precision of new gravity measurements is under ±0.01mGal and obtained elevation errors of differential GPS data is under ±0.5 m. The data were corrected to obtain the Complete Bouguer Anomaly (CBA) using standard correction processes (Blakely, 1995; Lowrie, 2007): tide correction, instrumental drift correction using daily repetitions at base stations, normal gravity correction, Free-Air, Bouguer, and Terrain corrections. These processes were conducted considering a 2.67 gr/cm3 reduction density. Earth tide correction was removed from the new data according to Longman (1959) algorithm. Normal gravity correction of new data considered the subtraction of the theoretical gravity of the WGS-84 ellipsoid. Free-air correction of all onshore data was calculated as 0.3086h (mGal), where h is ellipsoidal height in meters (Lowrie, 2007). Due to the inhomogeneities in the elevation measurement techniques used in old onshore data acquisition (registered between 1982 and 2002, Schmidt and Götze, 2006), we prefer to use SRTM elevation data to perform the Free-air and Bouguer corrections of these old gravity data. The terrain correction of all data was calculated following a combination of the algorithms proposed by Kane (1962) and Nagy (1966) and with high resolution SRTM elevation grid. The terrain correction includes topographic data located up to ~300 km around each station. GPS data processing, gravity data processing and all figures presented here consider geographic coordinates in the datum WGS84 and WGS84-18S for UTM coordinates.

The spatial coverages of different gravity databases (satellite, marine, and onshore) present areas of interception (Fig. 2) where they can be compared to determine the average gravity differences (constant average shifts). These shifts were used to generate a merged database levelling all data to the values observed in the new acquired data. The Free-air values of the onshore stations were used to move the Free-air anomaly of Sandwell and Smith (satellite data) to the same level (adding a shift of -17.3 mGal to Sandwell and Smith data), and finally GEODAS Free-air data (marine lines) were levelled and merged with the other data (adding a shift of -24.78 mGal to GEODAS data) to calculate the CBA (Fig. 3).

### 3.2 Density modelling

### 3.2.1 2D regional forward gravity models

In order to study the regional structure of the continental wedge and subduction zone, we modelled five profiles (P1_Toltén, P2_Unión, P3_Osorno, P4_Llanquihue and P5_Chepu, see Fig.2), which runs perpendicular to trench at latitudes of 39.25ºS, 40.2ºS, 40.5ºS, 41ºS and 42ºS, respectively. These profiles were extracted from the regional Complete Bouguer Anomaly grid obtained from the merged gravity database (Fig. 3). 2D forward modelling was performed by using the GravGrad modelling scheme (Maksymowicz et al., 2015), allowing the calculation of the gravimetric response of a stack of layers with arbitrary shape. The densities inside each layer can be varied along the vertical and horizontal directions. As the gravity anomalies are not exclusively dependent of the density structure below each data (i.e., it should be modelled considering masses around the profile), the elevation (bathymetry/topography) in 2D modelling is an averaged elevation profile which includes data inside a ~ 40 km wide band around the profile (i.e., averaging the elevation to 20 km on each side of the gravity profile). A 40 km wide band is a reasonable assumption considering that the wavelengths of the CBA gravity anomalies along the profiles are mostly larger than ~40-50 km. On the other hand, this parameter is not critical for the obtained 2D model. In fact, a completely different value (e.g.,10 km wide) can be considered with minor modification in the resulting density model (see more details in supplementary material).

As a forward modelling procedure, GravGrad allows to the user the iterative modification of geometry and densities of all layers in the model to fit the observed gravity anomalies (CBA in this case). Section 3.3 and Fig. 2 describe the independent information used to constraint the slab geometry, continental Moho depths and sedimentary thickness at the CD basin. General density structure of the slab, continental plate and mantle, were based on seismic/seismological Vp models available at the zone, converted to density by the empirical Nafe–Drake transformation curve (Brocher et al. 2005).

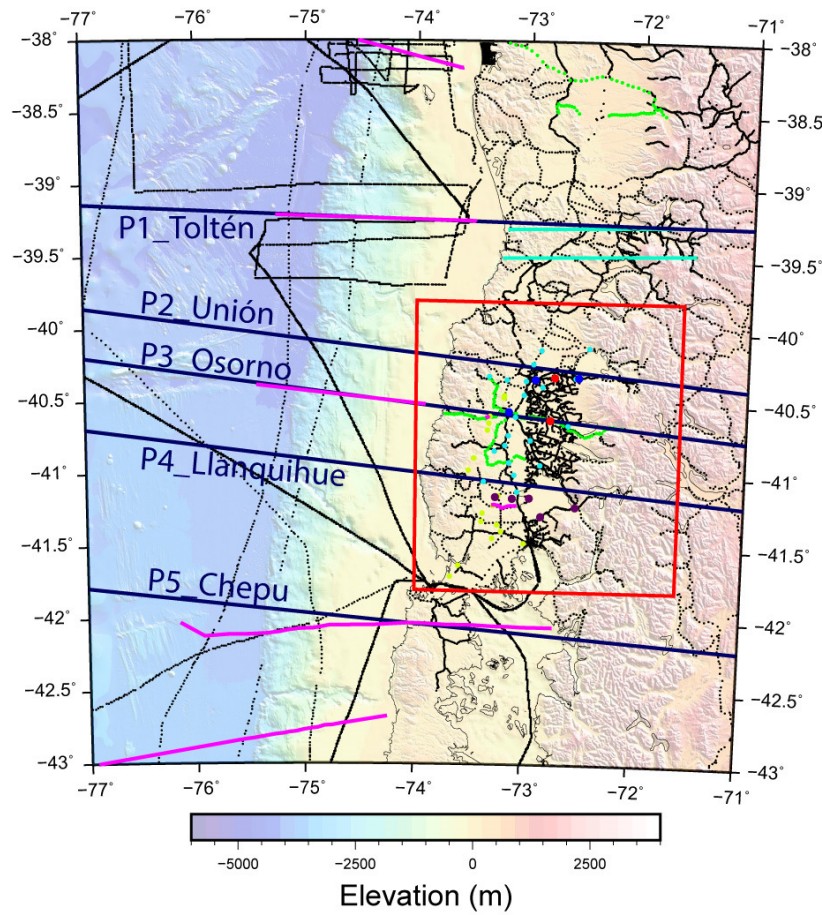

**Figure 2: Geophysical database in the study zone and the location of 2D and 3D models. The blue lines indicate the location of five 2D regional density model and the red rectangle is the zone in which local 3D inversion was obtained. The black dots designate gravity stations compiled by Schmidt and Götze (2006), onshore, and in GEODAS database (NOAA) offshore. The green dots illustrate the complementary gravity stations acquired by our group under the ANID-FONDECYT project Nº11170047. The blue and red dots**

**correspond to TEM soundings and MT stations acquired under the aforementioned FONDECYT project. Magenta dots indicate the MT stations presented by Segovia et al. (2021) and cyan dots correspond to TEM soundings published by DGA (2012). Receiver function profiles obtained by Dzierma et al., 2012a are shown with cyan lines. Location of seismic Vp-depth models (Contreras-Reyes et al., 2008; 2010; Bangs et al., 2020; Maksymowicz et al., 2021) and the seismic reflection lines presented by Jordan et al. (2001) and González et al. (1989) are indicated with magenta lines. The yellow dots correspond to the location of boreholes (McDonough et al.,**

**1997).**

### 3.2.2 3D gravity inversion

Regarding a more detailed analysis of the continental density structure onshore, a 3D inversion was performed in a central patch of study zone, where a large gravity maximum is observed paralleled to the Coastal Range (red polygon in Fig. 3). This onshore

3D density model was obtained using the UBC-GIF GRAV3D v3.0 software (Li and Oldenburg 1998). The algorithm inverts Residual Bouguer Anomaly (RBA) to derive a 3D density anomaly model of the crust. The Residual Bouguer Anomaly was generated by subtracting a 1th order polynomial trend from Complete Bouguer Anomaly data (see supplementary material). The 3D mesh has 67x80x102 blocks (in X, Y, Z direction, respectively). The horizontal mesh size is 3000 m x 3000 m. Due to the progressive sensitivity decrease of the Gravity inversion to sources in depth, UBC-GIF developers recommend using smaller

cells near the surface and increase the cell thickness with the model depth (https://www.eoas.ubc.ca/ubcgif/iag/index.htm). Accordingly, the cell size gradually grows from 100 to 1500 m in our model, reaching 70 km in depth.

A total of 3514 onshore gravity data points were used for the 3D inversion, generating a regular spaced grid (with a grid size of 3000 m x 3000 m). After numerous preliminary experiments, we set the length scale parameters of the UBC-GIF algorithm (Li and Oldenburg 1998) as 6000 m, 6000 m and 3000 m in X, Y and Z directions, respectively. These length scale parameters define the horizontal and vertical smoothness of the solution and preferred values are the double of the horizontal and vertical cell size used to discretize the media. This criterion is one of the recommended in the software manual (https://www.eoas.ubc.ca/ubcgif/iag/index.htm), but also it is important to highlight that sensitivity tests show that under a strong variation these parameters the obtained solution of density structure shows similar features (see details in supplementary material). The 3D inversion was constrained by information from the geological map of 1:1.000.000 scale (SERNAGEOMIN, 2003), MT stations, TEM stations and onshore seismic lines (Fig. 2). Accordingly, we performed the inversion with the following definitions for surface sediments and basement: a minimum homogeneous thickness of 500 m was assigned to the entire area of the model where Quaternary sediments are exposed. These cells can take densities between 1.9 gr/cm3 to 2.1 gr/cm3. The next 500 m (five cells) in depth correspond to a transition zone, where the blocks could be sediments or rock and can vary between 1.9 and 2.7 gr/cm3. The next 300 m (three cells) in depth correspond to a second transitory zone, where the blocks could be fractured rock or consolidated rock, they can vary between 2.4 and 3 gr/cm3. Below, the blocks correspond to basement can take values between 2.5 and 3 gr/cm3. Finally, below 7500 m depth we constrained the model to have greater densities than the background (2.67 gr/cm3). Then, those deep cells can take values between 2.67 to 3 gr/cm3 in order to ensure more realistic vertical gradients in the lower constrained deep portion of the model.

To include the presence of main lakes in the zone, the model is forced to be water in the blocks that correspond to lake, assigning them a density of 1 gr/cm3. The bathymetry of the 2 first lakes was obtained from Chilean National Oceanographic Service (http://www.shoa.cl/php/inicio) and in the case of Llanqihue Lake, a mean of 200 m of depth was considered. Similarly, to consider the gravimetric effect of the sea, the model was forced to be water in the blocks above bathymetry.

### 3.3 Geophysical constraints

### 3.3.1 Available geophysical information

Independent geophysical data was used to constraint the 2D density models and onshore 3D inversion (Fig. 2). This information includes: (1) The available 2D velocity-depth models at different latitudes (Contreras-Reyes et al., 2008; 2010; Bangs et al., 2020; Maksymowicz et al., 2021), used as a reference for general structure of the oceanic plate and marine continental wedge after Vp to density conversion according to the empirical Nafe–Drake transformation curve (Brocher et al. 2005). (2) Interpretation of reflection seismic profile (in depth) at ~42ºS (González et al., 1989). (3) The Quaternary sedimentary thickness and the top of the Paleozoic basement observed in the onshore ENAP seismic lines Z5B-010A and ZDO-001 (McDonough et al., 1997; Jordan et al., 2001) and ENAP boreholes (McDonough et al., 1997; Honores et al., 2015). (4) The SLAB 2.0 model (Hayes et al., 2018) to constrain the deep slab geometry. (5) Moho depth along the profiles presented by Dzierma et al. (2012a). (6) 1D electrical resistivity models using magnetotelluric measurements obtained by Segovia et al. (2021), 1D electrical resistivity models from TEM measurements presented by DGA (2012), and 1D electrical resistivity models from new MT and TEM measurements were used to constraint the thickness of young sedimentary fill at CD. At the MT/TEM stations where 1D resistivity models do not reach the base of the young sedimentary fill (by cultural electromagnetic noise or limited penetration in

thick sedimentary fill areas), we define values of minimum sedimentary thickness, aiming to decrease uncertainties in the density modelling.

### 3.3.2 Electromagnetic methods to constrain gravity measurements

The new magnetotelluric data (red dots in Fig. 2) were collected using Metronix ADU-08 data loggers and MFS-07 induction coil magnetometers along with Pb-PbCl electrodes. Time series data were recorded between 12 and 24h. All sites were processed using the robust method based on Egbert and Booker (1986). 1D resistivity models of new and previously measured data (from Segovia et al. 2021) were obtained using Occam (Constable et al. 1987) and Bostick (based on Bostick, 1977) algorithms

implemented in WinGLink (Schlumberger, version 2.21). See data and models in the supplementary material.

The Transient electromagnetics measurements (blue dots in Fig. 2) were carried out utilising the ABEM WalkTEM (ABEM, 2016). In general, a central loop setup was used with a transmitter loop size of 100 x 100 $m^2$ or 40 x 40 $m^2$.TEM stations were modelled by using Interpex-IX1D TEM software, generating 1D resistivity depth models by Ridge Regression algorithm (see

data and models in the supplementary material).

### 4 Results

### 4.1 Complete Bouguer gravity Anomaly (CBA)

It is necessary to describe the main characteristics observed in the CBA at the study zone (Fig. 3) before analysing density

models. The general aspect of the CBA is a sequence of bands with high and low gravity, roughly parallel to the margin. Offshore, we observe the low CBA associated to the deep trench seafloor and its sedimentary fill. It is important to notice that this low CBA extends several kilometres landward from the deformation front (toe of the continental wedge), which implies the presence of low-density units at the lower slope of the continental wedge. The main feature observed in the slope and shelf area is the low CBA zone (L1 in Fig. 3a) extended from ~38.5ºS to ~ 41.ºS, correlating whit a decrease of general slope angles at

same latitudes (Fig. 1). This morphologic and gravimetric anomaly is also correlated with the maximum slip patch of the giant Valdivia earthquake, as highlighted by several authors (Wells et al., 2003; Maksymowicz, 2015; Contreras-Reyes et al., 2017; see Fig. 3b).

Onshore, the regional aspect of CBA is an eastward trend of gravity decreases from the coast to PC, mostly related to the

280 presence of continental root below the Andes (Tassara et al., 2006; Tašárová, 2007). Therefore, high CBA anomalies are observed along the coast (Fig. 3a), but their amplitude decreases between ~38.7ºS and ~40ºS, where a relative low CBA anomaly is observed (L2 in Fig. 3a). It is necessary to consider that L2 is spatially correlated with a zone of landward extension of the CC and metamorphic complex outcrops (WS/ES).

To the east, a sequence of gravity lows, with sparse gravity maximums, is correlated with the eastern part of CD basin, the current volcanic arc and LOFS, suggesting a complex density structure at the PC zone. Between ~40ºS and ~41.5ºS (and probably southward) a prominent positive anomaly can be seen above the western portion of CD basin (H1 in Fig. 3a), indicating

the presence of high-density body elongated to the northeast and covered by sedimentary fill of CD. This interesting forearc gravity maximum was observed by Hackney et al. (2006) based on the same onshore data, and has been confirmed by our new complementary stations.

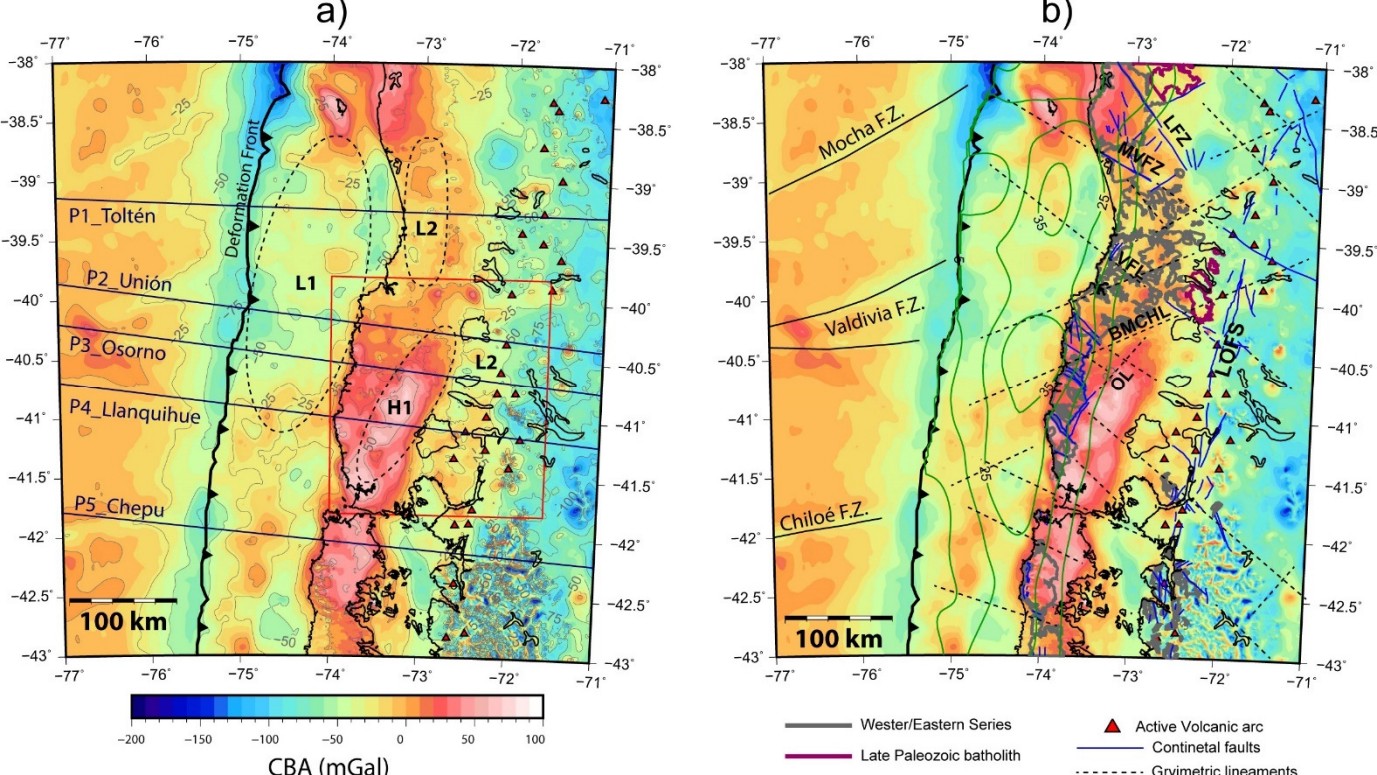

**Figure 3: Complete Bouguer Anomaly (CBA) at the study zone. a) Main gravimetric forearc features observed in the area (see main text for details). The blue lines illustrate the tracks of 2D forward density models and the red rectangle indicates the zone where 3D density inversion was carried out. Dotted black ellipses shows the approximated extension of anomalies L1, L2 and H1 along with other elements seen in Fig. 1a. b) Crustal tectonic structures and CBA. The figure includes the contours of metamorphic Western/Estern Series and Late Paleozoic batholith outcrops. Dotted black lines are interpreted gravimetric lineaments, e.g., Valdivia-Futrono lineament (VFL), Bahía Mansa-Choshuenco lineament (BMCHL) and Osorno lineament (OL). TheOther elements as in Fig. 1a.**

By a visual inspection of the CBA map, we interpret a set of gravity lineaments (Fig. 3b) with north-west and north-east directions. Derivative filters (directional derivatives, slope gradient and analytical signal) applied to CBA helped to identify these regional trends (see supplementary material). This qualitative interpretation confirms the location of fault zones previously identified at the surface (SERNAGEOMIN, 2003; Melnick and Echtler, 2006), suggesting their continuity through the forearc and, in some cases, their seaward extension (e.g., Valdivia-Futrono lineament, VFL in Fig. 3b). Additionally, new gravimetric lineaments are identified in CBA, suggesting the presence of large structures affecting the basement units (e.g., Bahía Mansa-Choshuenco lineament, BMCHL in Fig. 3b). H1 anomaly is limited to the north-west by the Osorno lineament (OL in Fig. 3b) which presents continuity with an identified west-dipping reverse faults in the south-west (SERNAGEOMIN, 2003; Melnick and Echtler, 2006; Hackney et al., 2006; Encinas et al. 2021), indicating that the geometry of H1 has a structural/tectonic control.

## 4.2 2D density profiles

Fig. 4 shows the results of the 2D forward gravity models obtained through the five studied profiles. As observed in Fig. 4a, the modelling process allowed to attain a good fit between observed and calculated gravity data, associated to low RMS values ($\leq$ 4.0 mGal) in comparison with the total amplitude of gravity anomaly. According to these results, the marine structure of the overriding plate can be described as a general landward increase of density between deformation front (DF at the trench) and the coastal area, where it is possible to define at least two internal units (Fig. 4b to 4f): The first one corresponds to a frontal low density unit ($\rho <$ ~2.5 gr/cm3) of about 25-35 km wide, with a rapid landward horizontal density gradient. This frontal unit is roughly correlated with the lower slope of the continental wedge. The second unit is characterized by a lower landward horizontal density gradient and shows densities between ~2.5 gr/cm3 to ~2.8-2.9 gr/cm3. This middle wedge unit is extended from the lower slope to the coast by ~70 km at profiles P2 to P5 and is slightly wider (~90 km) at the northernmost profile P1_Toltén. Immediately below the seafloor, all profiles present marine forearc basins with variable thicknesses (< 5 km) and densities lower than ~2.3 gr/cm3. Few kilometres westward from the coast, continental wedge shows a transition to higher densities landward (higher than ~2.9-3.0 gr/cm3 in the deep portion of the crust). This transition can be described as a landward limit of the middle wedge unit and seems to have a west dipping geometry.

At the onshore, the upper portion of the continental forearc (the upper ~10-15 km) displays a sequence of low- and high-density zones. Below CC (and metamorphic complex outcrops), the shallow densities are generally higher than 2.5 gr/cm3 and downward. However, this region is not particularly dense. In fact, below the sedimentary fill of CD basin we observe a high-density maximum in the five 2D profiles (H1 in Fig. 4b to 4f). Then, the results confirm the presence of a high-density zone associated the high CBA anomaly already described (D1 in Fig. 3a), suggesting its prolongation to the north-east and south-west. Comparing the 2D profiles, we notice that D1 is progressively closer to the coast, southward from profile P2_Unión (~40ºS), i.e., D1 presents a north-east trend, as suggested before in the CBA description. It is important to note that the presence of D1 is clear in all profiles except for P2_Unión in which this density anomaly is slightly raised from a more homogenous model of the upper continental crust.

To the east of D1, all profiles show another high-density zone (D2 in Fig. 4b to 4f). It is important to consider that the large LOFS approximately correlates with the western limit of D2 in profiles P1_Toltén, with the eastern limit of D1 in profile P2_Unión and with the eastern border of D2 at profiles P4_LLanquihue and P5_Chepu, which suggests a structural relation between the deep geometry of the high-density anomalies (D1 and D2) and LOFS.

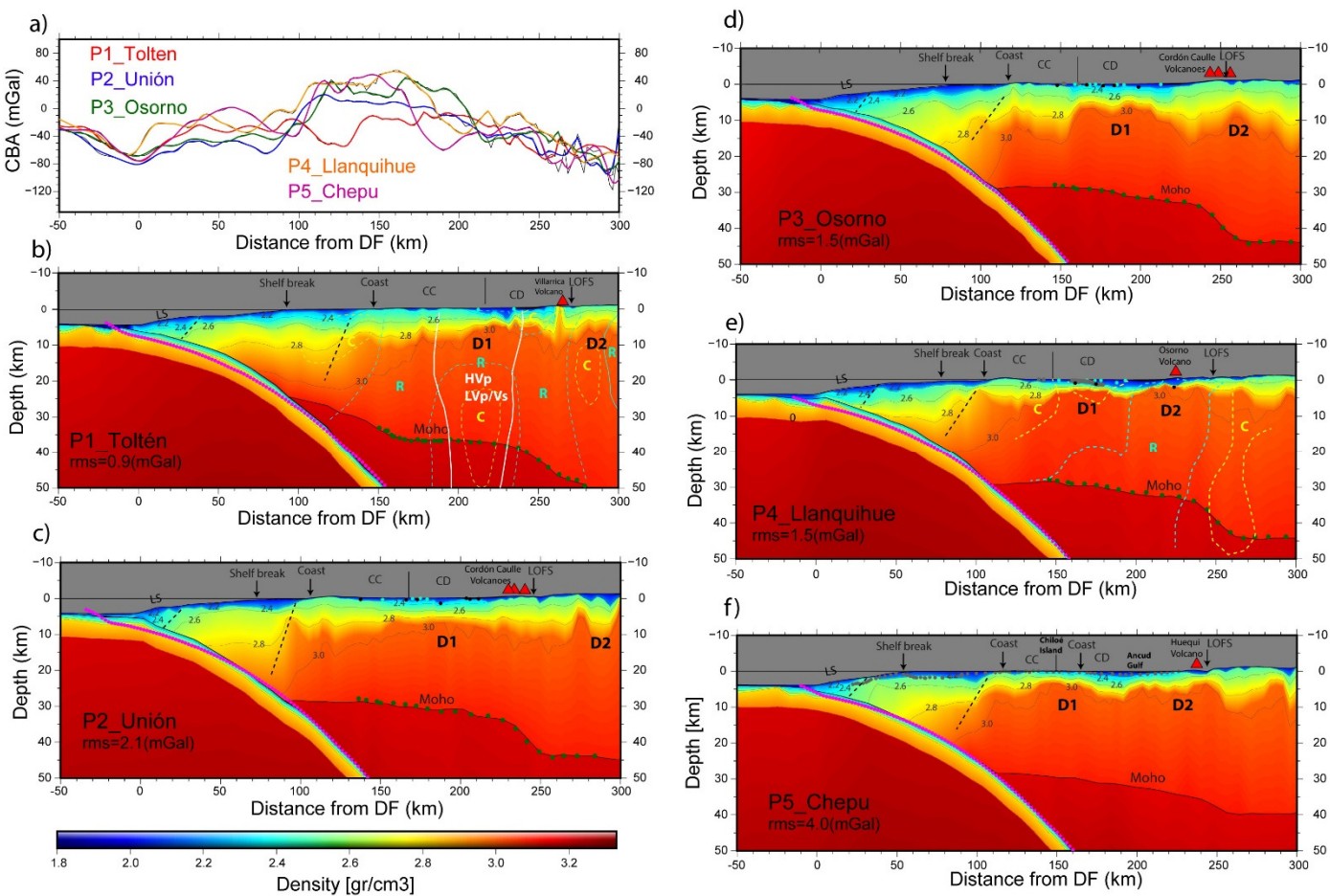

Figure 4: 2D regional forward models a) Complete Bouguer Anomaly (CBA) curves along five modelled profiles. Coloured curves are the modelled gravity signals and the corresponding observed data are presented in grey. b) Density depth model along the profile P1_Toltén, morphological limits of the continental wedge (as lower slope, LS) are indicated. Dotted black lines illustrate the approximate limits of frontal low-density unit and the middle wedge/shelf unit of the marine continental wedge. Thin vertical line indicates the limit between CC and CD. Red triangles correspond to the active volcanoes located near the profile. Pink dots correspond to slab geometry according to SLAB2.0 model (Hayes et al., 2018). Green dots depict the continental Moho depths obtained by receiver functions analysis (Dzierma et al., 2012a). Black dots indicate the base of poor compacted shallow sedimentary layer according to MT and TEM soundings and cyan dots correspond to minimum thickness of this sedimentary layer according to MT and TEM soundings. White lines limit a zone of high Vp and low Vp/Vs obtained by Dzierma et al. (2012b). Thin dotted yellow and cyan lines limit electrically conductive and resistive zones (C and R) according to Kapinos et al. (2016). c) Density depth model along the profile P2_Unión. d) Density depth model along the profile P3_Osorno. Grey dots show the base of poor compacted shallow sedimentary layer according to onshore seismic profiles and ENAP boreholes (McDonough et al., 1997; Jordan et al., 2001; Honores et al., 2015). Other elements as in a). e) Density depth model along the profile P4_Llanquigue. Thin dotted yellow and cyan lines limit electrically conductive and resistive zones (C and R) according to Segovia et al. (2021). Other elements as in d). f) Density depth model along the profile P5_Chepu. Grey dots represent the base of shallow sedimentary layers according to seismic reflection data (González, 1989) and other elements as in a). Individual figures of each profile are presented in the supplementary material.

## 4.3 3D local density model

As explained before, a large latitudinal change of onshore forearc continental structure is observed in the central profiles (P2_Unión, P3_Osorno and P4_Lannquihue), where D1 seems to have a north-east trend and where D2 is observed near the LOFS and the arc. This motivates the development of a local 3D density inversion in the forearc area, to derive the detailed structure of upper continental crust with and independent model strategy.

The 3D inversion modelled the input Residual Bouguer Anomaly (Fig. 5a) with high precision, as is observed in Fig. 5b, where

differences between modelled and observed data are, in general lower than ± 1 mGal. The results show density contrast

anomalies to about 20 km depth (Fig. 6, 7 and supplementary material), which means that deeper anomalies are mostly

contributing to regional linear trend of the CBA at the scale of 3D local inversion.

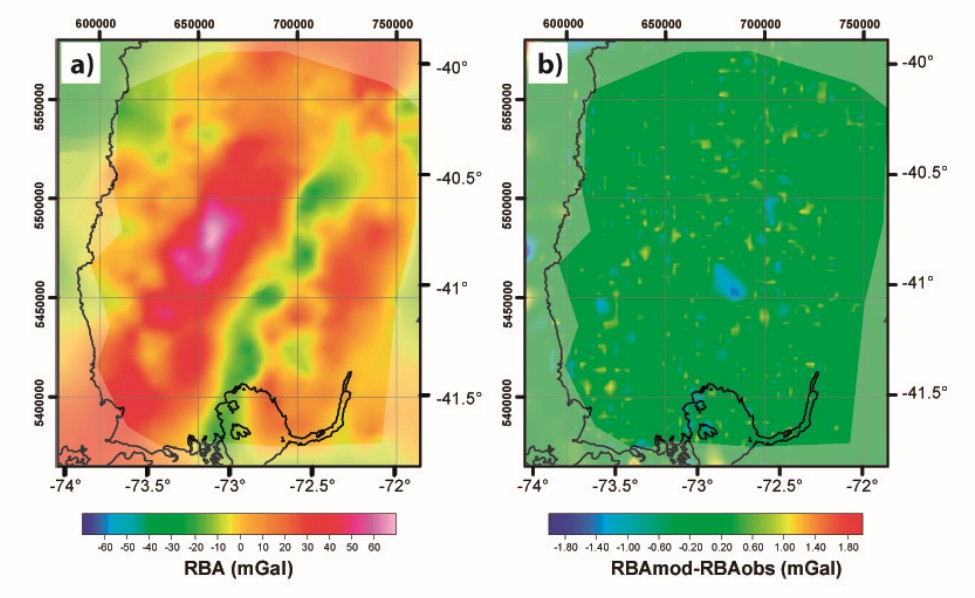


**Figure 5: a) Residual Bouguer anomaly (RBA) used as input for 3D inversion b) Difference between final modelled and input RBA data.**

Four constant depth slices through the final 3D model are shown in Fig. 6. In the slice at 49 m below sea level (Fig. 6a) it is

possible to observe the geometry of the CD basin as yellow zones that correspond to densities contrast between -0.77 gr/cm3 and

-0.57 gr/cm3. Also, it is possible to observe the areas that correspond to seawater as blue zone with densities contrast of -1.67

gr/cm3 and sedimentary areas below the lakes with densities contrast about -0.73 gr/cm3. Similar density structures are observed

in Fig.6b (at 1049 m below sea level).

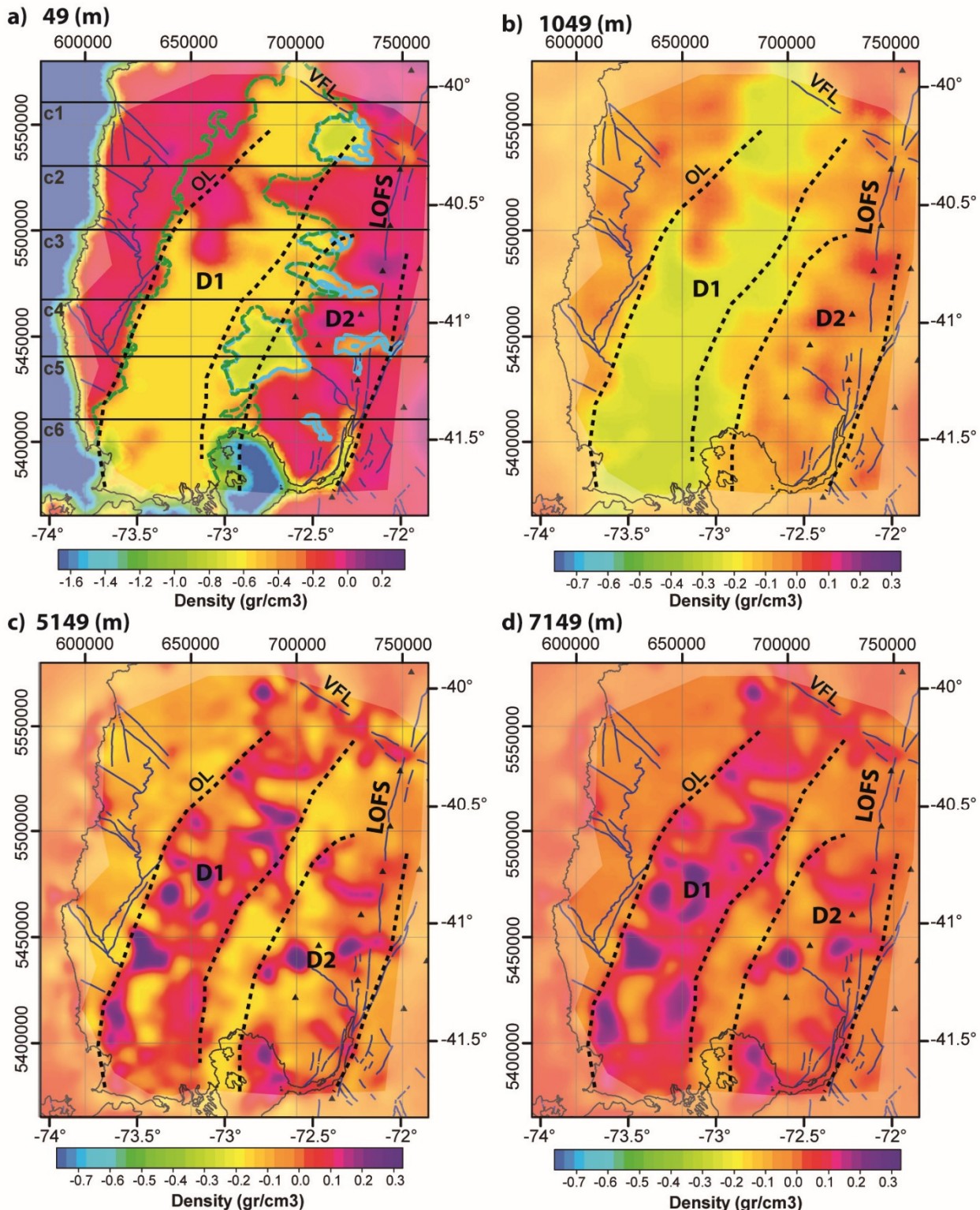

**Figure 6: Density slices from the 3D density model at different depths. a) Slice at 49 m below the sea level. Blue lines correspond to continental structures identified at the surface (SERNAGEOMIN, 2003; Melnick and Echtler, 2006), VFL highlights the Valdivia-Futrono lineament and OL corresponds to Osorno lineament (See Fig. 3b). Segmented green line illustrates the border of CD and the external limit of the lakes are highlighted with a cyan line. Active volcanoes are marked by black triangles. Segmented black limes indicate the approximate borders of H1 and H2 density anomalies (see main text for details) defined at deeper slice showed in d). Partially hidden zones are outside of onshore data considered for 3D inversion. b) Slice at 1049 m below the sea level. Note the change of colour scale in relation to a). Other elements as in a). c) Slice at 5149 m below the sea level. Other elements as in b). d) Slice at 7149 m below the sea level and other elements as in b).**



Fig. 6c and 6d, shows slices at 5149 m and 7149 m below sea level, respectively. In this figure it is feasible to observe the prominent high-density zone D1 under the CD, being consistent with the observed one in CBA and 2D models (Fig. 3 and 4). In the 3D model, D1 covers ~230 km along the strike and ~80 km in the horizontal direction, being oriented ~N25°E to the same direction of the western border of CC. The density contrast of this structure is higher than 0 gr/cm3 in most of the areas, reaching

0.3 gr/cm3 at denser zones. South of ~40.75ºS, D1 limits with WS outcrops to the west, while to the north-west it is bordered by the OL lineament (see Fig. 3b). The northern limit of D1 correlates with the presence of VFL lineament. To the east, a low-density lineament in the same D1 direction, with density contrasts from -0.17 to 0 gr/cm3, can be seen. This low-density band is about 10 to 15 km wide and limits to the east with the high-density zone D2. The 3D density model shows variations inside D1 and D2, which are formed by high density zones (density contrast ≥ 0.3 gr/cm3) merged with lower density areas.


To analyse the density variation with depth, Fig. 7 presents six W-E vertical cross sections of the 3D model at different latitudes (c1 to c6 in Fig. 6a). All cross sections show that D1 is below the CD and correlated to the eastern limit of the CC and WS outcrops, confirming the results obtained by 2D regional modelling. The top of D1 anomaly is obtained around 5 km below the sea level, displaying a trend of deepening to the north (also suggested by 2D modelling). At lower scale, the geometry of D1 is

characterized by two lobes and its western and eastern borders seem to be tilted to east and west, respectively (segmented black lines in Fig. 7). Eastward, D2 is modelled in the southern region at profiles c4, c5 and c6 (Fig. 7d to 7f) and it is also characterized by two lobes. The western limit of this anomaly suggests an inclination to the east.

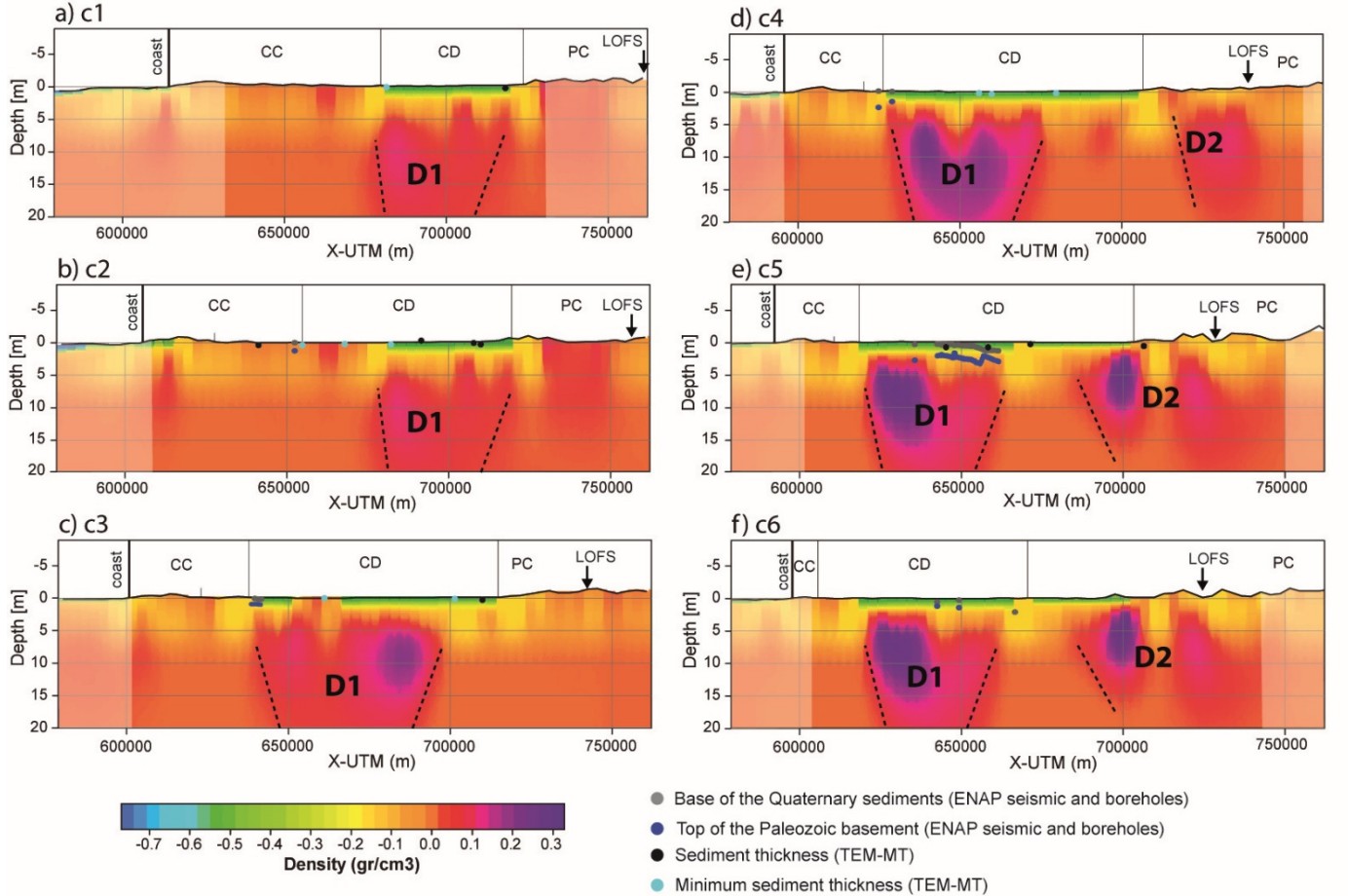

**Figure 7: Density-depth profiles extracted from 3D inversion model (see location in Fig. 6a). a) Profile c1 at UTM-North=5560500 m (WGS84-18S). Thin vertical lines indicate the limits of CC, CD. Black dots mark the base of shallow sedimentary unit, according to TEM/MT measurements and cyan dots correspond to the minimum sedimentary fill according to the TEM/MT soundings that do not reach basement. Grey and blue dots show respectively the base of Quaternary sediment and the top of Paleozoic basement according to onshore seismic profiles and ENAP boreholes (McDonough et al., 1997; Jordan et al., 2001; Honores et al., 2015). The interpretation of the approximate borders of the D1 anomaly is highlighted with dotted black lines. The partially hidden zones are outside of onshore data considered for 3D inversion. b) Profile c2 at UTM-North=5530500 m (WGS84-18S). c) Profile c3 at UTM-North=5500500 m (WGS84-18S). Grey and blue lines show the shallow sedimentary unit and the top of the metamorphic basement, according to seismic reflection profile ZDO-001 (see main text for details) and other elements as in b). d) Profile c4 at UTM-North=5467500 m (WGS84-18S). The approximate borders of the high-density anomalies D1 and D2 are highlighted with doted black lines e) Profile c5 at UTM-North=5440500 m (WGS84-18S). Grey and blues lines show the shallow sedimentary unit and top of the metamorphic basement, according to seismic reflection profile Z5B-010A (see main text for details) and other elements as in d). e) Profile c5 at UTM-North=5410500 m (WGS84-18S).**

## 5 Interpretations and discussions

These obtained results exhibit a landward segmentation of the continental wedge density structure observed from trench to arc (see an interpretative schema at Fig. 8a). Offshore, the frontal portion of continental wedge (to ~25-35 km landward from deformation front) presents low densities with a rapid horizontal increment of densities, interpreted as compaction process in the active accretionary prism along South-Central Chilean margin (Maksymowicz et al., 2015). It is also evidenced by seismic studies in the region (Moscoso et al., 2011; Tréhu et al., 2019; Contreras-Reyes et al., 2008; 2010; Bangs and Cande, 1997; Bangs et al., 2020). To the east, below the sedimentary fill of slope and shelf basins, the continental wedge is characterized by a second unit of higher density and lower horizontal density gradient (middle wedge unit, MWU). This unit can be associated to fractured basement rocks and/or more compacted units of a paleo-accretionary prism. In this sense, Contreras-Reyes et al. (2008) at ~38ºS and Contreras-Reyes et al. (2010) at ~43ºS interpret this unit (in Vp-depth profiles) as a paleo-accretionary prims of an undetermined age between Mesozoic to Tertiary. On the other hand, at ~39ºS and ~40.5ºS, Bangs et al., (2020) suggest that Paleozoic-Early Mesozoic accretionary complex (WS/ES) can extend further seaward to eastern limit of active accretionary prism (seismic backstop), in accordance to the interpretation of marine seismic data (and boreholes) of González (1989) at ~42ºS. However, the exploration boreholes presented by González (1989) were drilled in the shelf basin area, therefore, do not provide direct information about the age of the continental basement in the western portion of MWU.

Landward from MWU, the next segment correlates on the surface with the morphostructural domain of CC and shows a density increase respecting to marine wedge, but lower densities compared to continental crust below the CD and PC. Then, this CC domain is clearly related to the Paleozoic-Early Mesozoic accretionary complexes (WS/ES) and their continuity to depth. Gravity modelling techniques does not define a downward limit of WS/ES (without independent deep constraints). Nevertheless, interpretations of seismic reflection data at ~38.25ºS (Krawczyk et al., 2006; Ramos et al, 2018) showed the downward prolongation of WS/ES reaching deep levels near continental Moho interface (~30km depth). As previously mentioned, the seaward limit of WS/ES is not defined by direct lithological observations; their presence beneath the shelf basin is confirmed by exploration boreholes (González, 1989). Thereafter, the relative rapid change in velocity associated to the transition between MWU and CC domain (dotted grey line in Fig. 8) is interpreted as a structural limit (rather than a lithological change of the basement). This structural limit is probably associated with the development of the shelf basin and a general seaward increase of fracturing within the continental wedge. This structural interpretation seems to be confirmed by Contreras-Reyes et at. (2008) at ~38.25ºS, where continental intraplate seismicity (located by Haberland et al., 2006) is aligned with this limit, as well as the

intraplate seismicity located at 39.5ºS by Dzierma et al. (2012c).

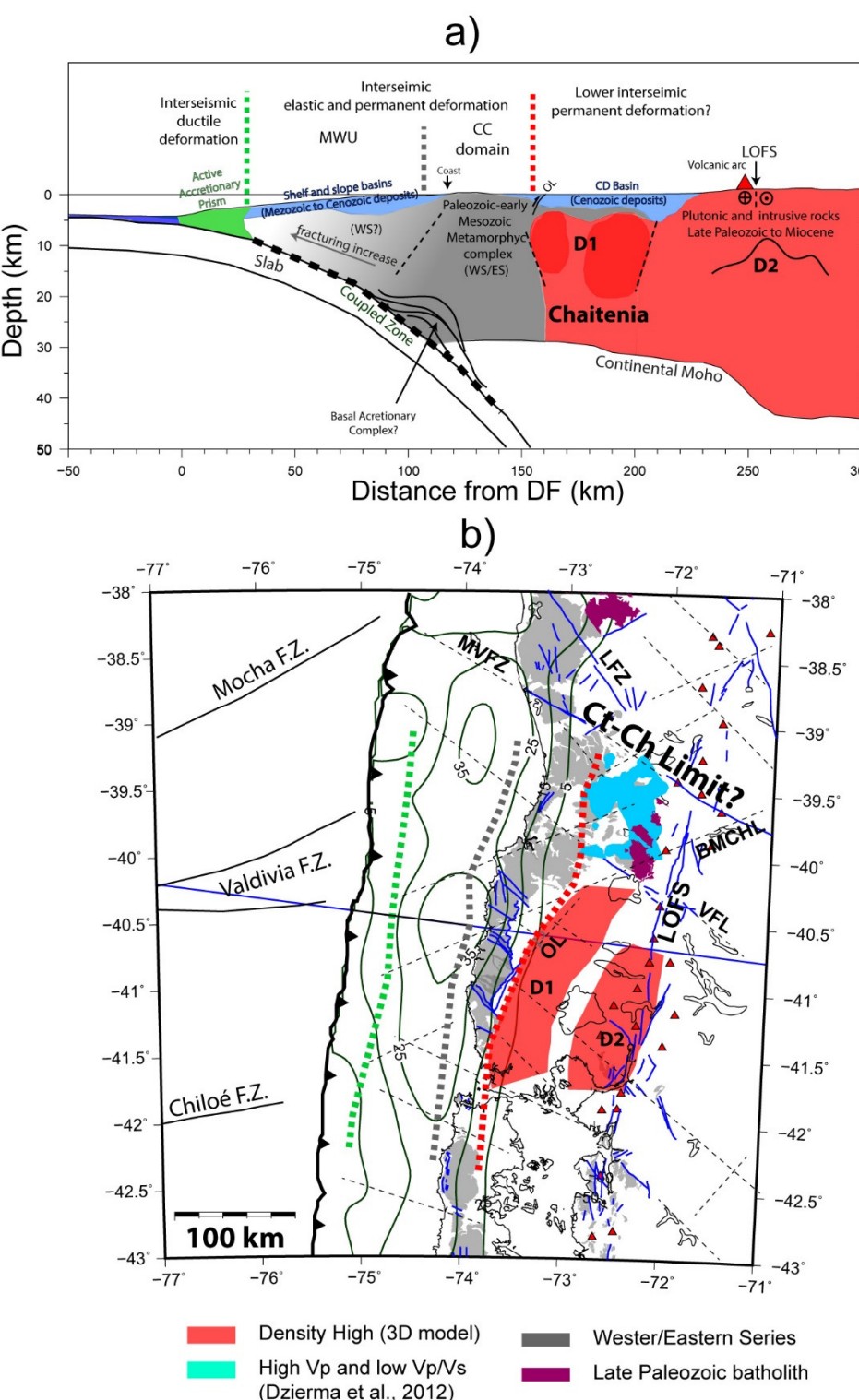


**Figure 8: Schematic interpretation of continental forearc structure at regional scale. a) Interpreted profile based on 2D density model along P3_Osorno line. Main geological/physical units are hatched with different colours. Light green for active accretionary prims, grey for Palaeozoic-Early Mesozoic metamorphic accretionary complex (WS/ES) and light red for high density continental crust interpreted as evidence of Chaitenia terrain in the forearc region. Dark red areas correspond to anomalies with high density contrast ( ≥ 0.1 gr/cm3) according to 3D model along the P3_Osorno line. Light blue indicates Mesozoic to Cenozoic deposits of slope and shelf basins and Cenozoic deposits in the CD basin. Segmented green, grey and red lines indicate the landward limits of active accretionary prism, Middle wedge unit and CC domain (see main text for details). b) Interpretation map of continental forearc structure. Light red areas highlight the high-density zones (H1 and H2) defined at ~7 km depth in the 3D density model (Fig. 6) and cyan areas correspond**


The CC domain extends landward to the contact with the D1 anomaly (dotted red line in Fig. 8). The eastern border of CC range at the surface correlates almost exactly with the western border of D1 in the 3D model (Fig. 6 and 7), which is also observed in 2D regional profiles (Fig. 4).  Accordingly, we understand that the continental crust of CC domain is deformed against a denser

(and probably more rigid) block of the continental crust observed here as D1 anomaly. As aforementioned, the lineament OL (Fig. 1b, 6 and 8) is continued to the south by a west dipping reverse fault that limits CC and CD (SERNAGEOMIN, 2003; Melnick and Echtler, 2006; Hackney et al., 2006; Encinas et al. 2021). This is an example of the contractional deformation styles that could be generated in the eastern border of CC by the depth contact between CC domain and D1. Interesting is to notice that the onshore refraction seismic profile ZDO-001, located to the west of D1 (in the CC along the profile P3_Osorno, see location

in Fig. 2) shows the inversion of an Oligo-Miocene normal fault, while the seismic profile Z5B-010A (located to the south of P4_Llanquihue profile) presents a minor contractional deformation in the CD sequences, above the D1 (Jordan et al., 2001). Deep below the CC, Maksymowicz et at. (2021) shows seismic reflectors at deep crustal levels with east dipping angles, which is consistent with the geometry of western border of D1 (Fig. 7) and supports a structural relation between the metamorphic complexes (WS/ES) and D1. However, the resolution of density model is not enough to calculate the precise inclination the

western border of D1, and this structural relation is only suggested by the results.

In the northern profile P1_Toltén, D1 correlates with the high Vp and Low Vp/Vs anomalies obtained by Dzierma et al. (2012b). Inside the region highlighted by white contour in Fig.4b, these authors show Vp/Vs values lower than 1.74, contrasting with values higher than 1.78, eastward and westward. In same region Vp and Vs models reach values ~4% and ~8% higher than

surrounding regions, respectively. Then, at least at the profile P1_Toltén, the correlation between D1 anomaly and the change in elastic properties is clearly observed.  Considering this Vs velocity anomaly and an increase of density of about 0.05 gr/cm3 (associated to D1 anomaly, Fig4a and 7), we estimated an increase of shear modulus at the order of 20% in comparison to the surrounding regions (at the same depth). To the south, this seismic velocity anomaly shows a clear continuity with the D1 geometry noticed in the 3D density model (Fig. 8b). This continuity indicates that D1 is a primary characteristic of the

continental crust, southward from ~39ºS, and this supports the interpretation of D1 as a dense-rigid zone.  The latitudinal analysis of these independent geophysical models establishes that those basement units associated with D1 are progressively shifted to east (and taken away from the trench), northward from 41.5ºS (Fig. 8b). In other words, the portion of the continental wedge formed by MWU and CC domain is ~50 km wider at 39.5ºS compared to the observed at 42ºS.

Outcrops of Late Paleozoic Batholith near ~40 ºS, are observed in the western border of WS/ES and D1 anomaly, possible implying that D1 is the southward continuation of the Late Paleozoic Batholith. However, outcrops of this Batholith are described at ~40.3ºS (Deckart et al., 2014) and ~42.5ºS (SERNAGEOMIN, 2003) near to volcanic arc, indicating a possible association of this unit with D2 anomaly. In this case, south of 39ºS, D1 should be a high density basement unit located westward from Late Paleozoic (Pennsylvanian) Batholith. An interesting candidate to fit these conditions is Chaitenia terrain (Ct in Fig. 1b)

which is described as an island arc, accreted to the Gondwana margin during late Devonian time (Hervé et al., 2016; 2018). The northward limit of D1 (high Vp and Low Vp/Vs anomaly, Dzierma et al., 2012b) can be roughly defined by the MVFZ. This structure could be interpreted as a limit between Chaitenia and Chilenia terrain to the north (Fig. 8b). This interpretation raises an interesting question about the role Chaitenia/Chilenia limit in the observed westward shift of the Late Paleozoic Batholith

southward of 38ºS and its relationship with the continental deformation generated by the kinematics of LOFS (Cembrano et al. 1996; Melnick et al., 2009; Geersen et al., 2011). On the other hand, Plissard et al. (2019) observed that outcrops of mafic and ultramafic (serpentinites) rocks associated to WS (south of 39ºS) shows P-T patterns (and structural characteristics) which that allow interpreting these units as rock located below an incipient back-arc basin during Devonian times (380-370Ma), which were incorporated to the subduction channel, reaching depth to about 60 km downward in the interpolate boundary (during Carboniferous time), to be finally exhumed in the eastern border of accretionary wedge during Permian time. Under this interpretation the eastern border of Devonian island arc (associated to Chaitenia terrain by Hervé et al., 2016; 2018) corresponds to an incipient back-arc, rather than a subduction zone, but the process finally ends in the accretion of Devonian island arc to the Gondwana margin. Again, these accreted units could be related to D1 anomaly in the region.

Beyond the lithology and age of D1 and D2 anomalies it is necessary to highlight the spatial association between the active volcanism and the main lineaments of the LOFS (Lara and Folguera, 2006; Sánchez et al., 2013; Díaz et al., 2020). Fig. 6 shows that most of the quaternary volcanoes are located above the local regions of relatively low-density contrast (in general < 0.0 gr/cm3 below 5 km depth) inside D2. This local 3D pattern of density anomalies is not easy to interpret in 2D regional models, because they are averaging the density structure around the profiles and shows that the 3D local inversion is a relevant methodology (complementary to 2D regional analysis) to observe medium depth and shallow density structures in the upper crust. These local regions of relatively low-density contrast could respond to more fractured regions of the upper crust as a response of deep structures associated to branches of LOFS and other continental structures presents below the CC and CD. In fact, elatively low-density zones may be related to active volcanic processes observed along this fault system in the Araucanía, Los Ríos and Los Lagos districts. As shown by Díaz et al. (2020), relatively low electric resistivity is found at depths between 7 and 15 km below the local trace of the LOFS, east of Osorno volcano, associated in this case with a zone of partial melt related to a deeper ascent of basaltic magmas enhanced by the LOFS, and therefore a lower density compared to its surroundings.

The upward migration of magmas should generate local weakening zones in the overriding plate, and consequently, the continental crust in the active volcanic zone should present pervasive fracturing, fluid migration and lower density. Hence, the basement extended to the east of D1 could correspond to a similar lithology but affected by the pervasive fracturing and fluid migration processes associated to active volcanic arc and LOFS. This interpretation is supported by the increase of Vp/Vs values to the east of D1 (Dzierma et al., 2012b), at least at the profile P1_Toltén (Fig 4a). On the other hand, Kapinos et al. (2016) and Segovia el al. (2021) describe electrically conductive anomalies eastward from LOFS and high resistivity values beneath the CD basin (Fig. 4b and 4e). To the west of D1, these authors also establish conductive anomalies associated with CC domain, reinforcing the interpretation of a transition from highly deformed and fractured basement (related to deep units of WS/ES) to a denser/rigid basement below CD.

It is already known that the rupture propagation during large earthquakes, the interseismic deformation (including aftershocks and foreshocks), as well as the interplate locking are complex processes that depend primary on the frictional properties at interplate boundary (subduction channel) and the stress field evolution (Scholz, 1998; Perfettini, H., and Avouac, 2004; Tassara, 2010; Moreno et al., 2018; Im et al., 2020). As aforementioned, the segment of continental wedge that includes MWU and CC domains, i.e. fractured and/or metamorphic basement units, is progressively wider to the north of 42ºS. This structural change correlates with the patch of high coseismic slip of 1960 Mw9.6 Valdivia earthquake (Fig. 8b), which added to the correlation with gravity anomaly L1(Fig. 3a) and with changes in slope morphology, suggest a link between the megathrust seismotectonics

and physical properties of the overriding plate. In this regard, we propose that the MWU and CC domains correspond to a portion of the continental plate displaying a higher elastic and permanent deformation compared to the rigid basement landward (Chaitenia/Chilenia). Consequently, the change in the horizontal extension of this unit should modify the process of stress loading during the interseismic periods.

Due to the relatively scarce seismological data in the area and the long recurrence time for large events, it is difficult to conceptualize the complete seismotectonic story of the study zone. Nevertheless, some observations seem to support our hypothesis. Firstly, the rupture zone and aftershocks (including continental intraplate events) of the Mw 7.6 earthquake occurred in 2016 at 43.5ºS. This event was the largest since 1960 in the rupture area of Valdivia earthquake (Moreno et al., 2018; Lange et al., 2018) were located at the base and within the CC domain, in the western border of a high Vp-low Vp/Vs anomaly (Lange, 2008). This velocity anomaly is a clear continuation of D1 to the south of the studied area. On the other hand, historical (not instrumentally recorded) megathrusts events activated this segment of the margin in 1737 (Mw~7.5) and 1837 (Mw~8). They have been associated to ruptures extended to the south of ~39ºS (Kelleher, 1974; Lomnitz 2004), indicating that the northern portion of 1960 Valdivia earthquake could have different mechanical properties.

Lithology and internal deformation style inside and at the base of MWU and CC domains can play a different but complementary physical role on the seismotectonic segmentation of the margin. The high slip patch of Valdivia earthquake also correlates with the segment where the geometry of marine continental wedge (seaward from shelf break) is consistent with a decrease in the effective friction coefficient ($\mu_b$*) at the interplate boundary (Maksymowicz et al., 2015). This suggests oversaturate fluid conditions in the subduction channel, at least in the western portion MWU at the study zone. At the same time, according to Menant et al. (2019), the deformation style of basal accretionary complexes (typically an antiformal stack of duplexes) favoured upward fluid fluxes from the interplate boundary, generating dewatering and the increase of $\mu_b$* in some adjacent regions of the subduction channel (mainly downward from basal accretionary complex). Several authors have suggested the presence of this deformation style in the deep zone of WS unit (Krawczyk et al., 2006; Ramos et al, 2018, Moreno et al. 2018; Maksymowicz et al., 2021). Under this interpretation, the widening of MWU and CC domains to the north of ~42ºS could favoured high friction in the deep region of interplate boundary (bellow CC domain) and a relatively low friction in the seaward portion of MWU. Therefore, the position and horizontal extension of the WS could be linked to changes in the frictional properties along the megathrust. However, more studies should be done to explore the seaward limit of WS/ES and the internal structure of MWU and CC domain.

**6 Conclusions**

 2D and 3D density models of the forearc show a landward and latitudinal segmentation of the continental wedge in the studied zone. Offshore, the active accretionary prism limit with a more competent basement below the middle wedge and shelf, which exhibits a landward increase of density, probably associated with a progressive decrease of fracturing. To the east, the Coastal Cordillera domain presents an increase in the upper crust densities but reaching lower values than the observed in the high-density anomaly below the Central Depression. Northward from ~42ºS, this high-density anomaly is seen progressively further from the trench, determining a northward widening of the middle wedge and Coastal Cordillera. This feature correlates with the

high slip patch of the giant 1960 Mw9.6 Valdivia earthquake

Based on geological information, we associate the middle wedge unit (at least its eastern portion) and Coastal Cordillera domain with the Late Paleozoic-Early Mesozoic accretionary complex, and the high-density anomaly below the Central Depression as a geophysical evidence of Chaitenia terrain. The deformation style at the eastern border of the Costal Cordillera and seismological studies support the hypothesis of a more rigid behaviour of the continental crust below the Central Depression. Accordingly, we propose that changes in the horizontal extension of the middle wedge unit and Coastal Cordillera domain should have modified the process of stress loading during the interseismic periods, and that changes in position and extension of the Late Paleozoic-Early Mesozoic accretionary complex could be linked to the frictional properties of the interplate boundary.

Our results highlight the role of the overriding plate structure on the seismotectonics process in subduction zones, but more studies are necessary to understand the changes in physical properties (elasticity, temperature, among others) associated with the geological story of the margin. This work motivates similar analysis of the continental basement in other subduction margins, as in the 2010 Mw8.8 Maule earthquake and Mw9.0 Tohoku-oki ruptures zones.

### Acknowledgments

This work was funded by CONICYT/ANID under the Chilean Fondo Nacional de Desarrollo Científico y Tecnológico (FONDECYT), grant 11170047. We also thank the support of FONDECYT 1211257 and CONICYT/ANID- PIA/Anillo de Investigación en Ciencia y Tecnología ACT172002 project "The interplay between subduction processes and natural disasters in Chile".

### Author contribution

AM designed the gravity experiment and DD designed MT and TEM experiments. Data acquisition was performed by AM, DMC and DD. AM and DMC developed the gravity data processing and 2D/3D models. MJS and DMC performed MT/TEM data processing and modelling. The interpretation and discussion were developed by AM, DMC, DD and TR. AM prepared the manuscript with contributions from all co-authors.

### Competing interests

The authors declare that they have no conflict of interest

### Code/Data availability

Data, models and GravGrad routines are available at Maksymowicz, A. (2021, April 29). Forearc density structure of the overriding plate in the northern area of the giant 1960 Valdivia earthquake. Retrieved from osf.io/y9aph

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
