# Peer review of "Forearc density structure of the overriding plate in the northern area of the giant 1960 Valdivia earthquake."

_Solid Earth, 2021_

## Referee Comment (RC1)

**Review for "Forearc density structure of the overriding plate in the northern area of the giant 1960 Valdivia earthquake"**

In this study, the authors have explored the continental fore-arc density structure of the Nazca-South America subduction zone. They compile a gravimetric data base combining public databases and lots of new measurements. They perform 2D and 3D inversion for a detailed density structure both onshore and offshore and further calibrated the results with 1D electrical resistivity models from MT and TEM measurements. The authors show the spatial distribution of the mantle wedge and the Coastal Cordillera domain resolved from this density structure. They propose a model for the current stress and friction evolution on the subduction plate which may relate to the high slip patch distribution of the giant 1960 Mw9.6 Valdivia earthquake. The study provides the density structure along the coast in detail and a geophysical perspective to understand the subduction process.

This study covers an activate region where huge megathrust earthquakes have taken place. The results are significant for understanding the environment of megathrust earthquakes and thus important for future hazard assessment. The manuscript is well written. However, I found some missing references for the method and data process, see comments below. Thus I recommend it to be published in *Solid Earth* with minor revision.

Minor comments:

Figure 1a. This is a very nice figure but with too many colors. I suggest to use gray scale for elevation.

Line 71: This is an old reference. Maybe use MORVEL.

Line 135: a reference of Lacoste & Romberg is missed.

Line 140: it is not clear how is this reference density used.

Line 150: why use 40 km-wide band? Any test on this?

Line 172: why does the cell size change along depth?

Line 176: why choose 6000 m by 6000 m by 3000 m? Additional information is needed to support this.

Line 216: more explanations for WinGLink.

Figure 4, 5,6,7: I feel some labels are too small. Please use consistent font size.

Line 323: how is this initial input model chosen?

Line 365: how is the dashed line of the incline of H1 determined? Here some explanations are needed. I am also wondering if the angle H1 has some implication of the subduction structure.

Line 386: is this compact process supported by other studies? Geological or geochemical?

Line 390: what is the physical meaning of this low horizontal gravity gradient?

Line 404: how well is the depth (30 km) of WS/ES constrained?

Line 410: it is interesting that the depth limit is related to seismicity. Is this observed in other subduction fault?

Line 460: Normally the fluid migration can be associated with high Vp/Vs ratio in subduction zone. Is there Vp/Vs ratio variation that is associated with the fluid migration here?

---

## Referee Comment (RC3)

[referee-annotated manuscript omitted]

---

## Author Comment (AC1)

**Response to comments of the anonymous reviewer on the manuscript "Forearc density structure of the overriding plate in the northern area of the giant 1960 Valdivia earthquake" (se-2021-53)**

First, we gratefully thank the comments on our work from the anonymous reviewer (https://doi.org/10.5194/se-2021-53-RC1). In this document we write responses in blue after the reviewer's comments (black). The corresponding changes in the text were included in the new submitted version of the manuscript.

Sincerely

The authors

Review for "Forearc density structure of the overriding plate in the northern

area of the giant 1960 Valdivia earthquake"

In this study, the authors have explored the continental fore-arc density structure of the Nazca-South America subduction zone. They compile a gravimetric data base combining public databases and lots of new measurements. They perform 2D and 3D inversion for a detailed density structure both onshore and offshore and further calibrated the results with 1D electrical resistivity models from MT and TEM measurements. The authors show the spatial distribution of the mantle wedge and the Coastal Cordillera domain resolved from this density structure. They propose a model for the current stress and friction evolution on the subduction plate which may relate to the high slip patch distribution of the giant 1960 Mw9.6 Valdivia earthquake. The study provides the density structure along the coast in detail and a geophysical perspective to understand the subduction process.

This study covers an activate region where huge megathrust earthquakes have taken place. The results are significant for understanding the environment of megathrust earthquakes and thus important for future hazard assessment. The manuscript is well written. However, I found some missing references for the method and data process, see comments below. Thus I recommend it to be published in Solid Earth with minor revision.

Minor comments:

Figure 1a. This is a very nice figure but with too many colors. I suggest to use gray scale for elevation.

R.- In the new version of the manuscript use a gray scale for elevation in Figure 1a.

Line 71: This is an old reference. Maybe use MORVEL.

R.- In the new version we referenced to Kendrik et al. (2003) and Vigny et al. (2009). According to DeMets et al. (2010), MORVEL have differences with GPS observations (it gives ~7.4 cm/yr in comparison to ~6.6 cm/yr).

Line 140: it is not clear how is this reference density used.

R.- We replace "reference density" by "reduction density" for more clarity. Both are referring to the constant density used in the standard gravity reduction process to obtain the complete bouguer anomalies.

Line 150: why use 40 km-wide band? Any test on this?

R.- In order to represent the regional density structure in a 2D model, some assumptions should be done. In particular, the infinite extension of the model in the direction perpendicular to profile overestimates the effect of local topographic features located exactly in the profile track. To avoid this problem, we averaged the topography (in the direction perpendicular to profile) in a 40 km-wide band (i.e., 20 km to both sides of the line). A 40 km-wide band is a reasonable assumption considering that the wavelengths of the CBA gravity anomalies along the profiles are mostly larger than ~40-50 km as is showed in the amplitude spectrum (see figure below). Then, the averaging of topography at this scale is an appropriate option.

Amplitude spectrum of BCA signal along the 5 gravity profiles used to generate 2D forward models. Dotted vertical line highlights the wavelength equal to 40 km.

On the other hand, the direct comparison of the results obtained with completely different value (10 km-wide, i.e. 5 km to both sides of the line) can be considered with minor modification on the resulting density model, which indicates that this "wide" is not a critical value for our modeling scale (see figure below for Profile Osorno).

Comparison between 2D forward models obtained with different bands of topography-averaging in the Osorno profile. Central panel corresponds to model obtained with a band 40km-wide of topography averaging (corresponding modeled gravity is presented in blue at the upper panel). Lower panel corresponds to model obtained with a band 10km-wide of topography averaging (corresponding modeled gravity is presented in red at the upper panel. Observed data corresponds to green line in the upper panel).

This analysis was mentioned in the new version of the manuscript and the figures were included in the supplementary material.

**Line 172: why does the cell size change along depth?**

R.- Due to the progressive sensitivity decrease of the Gravity inversion to sources in depth, UBC-GIF developers recommend using smaller cells near the surface and increase the cell thickness with the model depth (https://www.eoas.ubc.ca/ubcgif/iag/sftwrdocs/technotes/faq.htm). Accordingly, the cell size gradually grows from 100 to 1500 m in our model.

Line 176: why choose 6000 m by 6000 m by 3000 m? Additional information is needed to support this.

R.- In UBC-GIF software, length scale parameters define the horizontal and vertical smoothness of the solution (https://www.eoas.ubc.ca/ubcgif/iag/index.htm). We preferred values of LE =6000 m, LN =6000m and Lz =3000m, which is the double of the horizontal and vertical cell size (maximum vertical cell size=1500 m) used to discretize the media. This criterion is one of the recommended in the software manual (https://www.eoas.ubc.ca/ubcgif/iag/index.htm), but also it is important to highlight that sensitivity tests show

that, under a strong variation these parameters, the obtained solution of density structure shows similar features. This point was included in the new version of the text supported by figures in the new version of supplementary material.

Line 216: more explanations for WinGLink.

R.- WinGLink is a commercial software developed by Schlumberger, widely used in the industry and scientific community as well, with different modules to process, model and interpret electromagnetic data. Here we used the 1-D inversion module, based on Occam and Bostick procedures, described in the references given in the manuscript.

Figure 4, 5,6,7: I feel some labels are too small. Please use consistent font size.

R.- In the new version of the figures we changed the font labels.

Line 323: how is this initial input model chosen?

R.- As is explained in the text, the inverted input data is the Residual Bouguer Anomaly obtaining by removing a regional (linear) trend from CBA data.

Line 365: how is the dashed line of the incline of H1 determined? Here some explanations are

needed. I am also wondering if the angle H1 has some implication of the subduction structure.

R.- We note in the that the western limit of this anomaly suggests an inclination to the east as is observed in the 3D model (Fig. 7), and then we include segmented lines to highlight this interpretation. In order to clarify this, we write the following description in the figure caption:

"The interpretation of the approximate borders of the D1 anomaly is highlighted with dotted black lines"

On the other hand, regarding a possible relation between the shape of D1 and subduction angle it is in interesting to note that deep seismic reflectors located by Maksymowicz et at. (2021) below the Coastal Cordillera (i.e., westward from D1 at deep crustal levels) present, in general, east dipping angles. Then, we can interpret that the western border of D1 have a shape consistent with the structure of the Metamorphic complexes (WS/ES) at depth. However, the resolution of gravity model is not enough to calculate the precise inclination angle, and this structural relation is only suggested by the results. This idea was included in the new version of the manuscript.

Note that H1 and H2 density anomalies are named as D1 and D2 in the new version

Line 386: is this compact process supported by other studies? Geological or geochemical?

R.- Currently, the structure, lithology and fracturing of the continental wedge is interpreted mainly from geophysical studies. Unfortunately, the few available drills in Chilean Margin reach only the upper portion of the crust (generally the first kilometer below the seafloor), and then, lateral variation of properties as compaction and/or fracturing is not confirmed by direct observations.

Line 390: what is the physical meaning of this low horizontal gravity gradient?

R.- We use the concept of "horizontal density gradient" as the variations of density along the profile at a specific depth. This variation highlights the lateral changes in elastic properties that can be related with lithology, compaction and/or fracturing.

Line 404: how well is the depth (30 km) of WS/ES constrained?

R.- As is mentioned in the text, the interpretation of the seismic reflection data (Krawczyk et al., 2006; Ramos et al, 2018) supports the continuity of WS/ES to depth, which is expected for units that were formed and exhumated by basal accretion. Seismic data locate the depth of reflectors with precisions at the order of few kilometers, but clearly the lithology of deep portion of the crust cannot be univocally determined from geophysical data.

Line 410: it is interesting that the depth limit is related to seismicity. Is this observed in other

subduction fault?

R.- In the text we discuss the published earthquake catalogs in the study zone, but there are some examples of a correlation between the upper plate seismicity and changes in the seismic velocities and/or density along the continental wedge. Between others, Contrereas-Reyes et al., (2015) and Comte et al., (2019) shows similar correlations in Central Chile and Petersen et al. (2021) at northern Chile.

Line 460: Normally the fluid migration can be associated with high Vp/Vs ratio in subduction

zone. Is there Vp/Vs ratio variation that is associated with the fluid migration here?

R.- We agree, in this paragraph we should highlight the Vp/Vs model. Variations in Vp/Vs are associated to variations in rigidity, which can be related to fluid content of the rocks, but also with porosity/fracturing and lithology. In this case the model of Dzierma et al (2012) shows a low Vp/Vs (and High Vp) region correlated with H1 at the northern profile P1\_Toltén. Then, we interpret this anomaly as a more rigid zone (and maybe less fractured/hydrated) compared to the surrounding crust.

---

## Author Comment (AC3)

**Response to comments pointed out by Andres Tassara on the manuscript "Forearc density structure of the overriding plate in the northern area of the giant 1960 Valdivia earthquake" (se-2021-53)**

We sincerely appreciate the detailed review and comments of the Dr. Andres Tassara (https://doi.org/10.5194/se-2021-53-RC2). To provide a complete response, we write all original comments in black and our responses in blue. The corresponding changes in the text were included in the new submitted version of the manuscript.

The authors

This paper presents a gravity-based model of the subduction zone anatomy along a segment of the Chilean margin that coincides with the greatest slip patch of the giant Valdivia 1960 earthquake. The authors attempt to interpret their resulting density model in terms of the geological structure of the margin and to discuss the implications of their findings for our comprehension of seismogenic processes leading to the largest earthquake ever recorded.

The work has scientific merit, although the density model is not particularly new (other papers have been published with the topic in the studied area) and the interpretations and conclusions will probably have a moderate impact in the community. This potential impact could be improved if a) more details are offered regarding some methodological aspects that are obscure in the present version (and therefore inhibit a clear interpretation of the results), b) the quality and clarity of figures and text could be augmented, and c) a deeper and complete interpretation can be developed. I expand points a and c below. I also provide a list of many idiomatic points (mostly orthographic and grammar typos and errors) that I identified through the text; this list is not complete, and the authors should take care and be sure that a revised version of the manuscript must have no errors like these in order to be accepted.

- Gaps in methodology.

I found that section 3 Data and Methods is oversimplified and some important gaps can be identified that you should fill in order to provide a better basis for the further interpretation of results. This is a list of my main concerns associated to line numbers on the original pdf.

135-136. Specify the model of the Lacoste&Romberg gravimeter used for the study. What do you mean with "with a digital upgrade provided by ANID-FONDECYT project No11170047"?

R.- We corrected as: "The gravity acquisition was made using A Lacoste & Romberg G-411 gravimeter, with a digital upgrade (http://www.gravitymeter-repair.com) funded by ANID-FONDECYT project Nº11170047"

134-140. More details about the processing of gravity data could be useful. How do you ensure that the new data are leveled with the old data compilation? Do you applied any further procedure to correctly merge the satellite data, marine data, old land data and your newly acquired data? Just putting all together as it was provided by different sources could create large problems with different data levels that must be solved before modeling!

R.-. The spatial coverages of different gravity databases (satellite, marine, and onshore) present areas of interception (Fig. 2) where can be compared to determine the average gravity differences (constant shifts). These shifts were used to generate a merged database levelling all data to the values observed in the new acquired data. More details were included in the new version of the manuscript.

171-173, 175-176. It seems that information provided in lines 171-173 regarding the size and geometry of the 3D inversion grid, is in contradiction with information provided in lines 175-176. Please clarify this point.

R.-. First paragraph refers to the mesh size and second paragraph refers to the "length scale parameters" of the UBC-GIF GRAV3D v3.0 software (Li and Oldenburg 1998). The point is clarified in the new version of the text.

Instead of gr/cc please use gr/cm3 in the entire text (which is the correct use of metric units in the SI)

R.-. changed in the new version.

You mention that "…greater densities than the background below 7500 m depth." What is the background density and its value?

R.-. The modeling of residual gravity provides contrasts of density respecting to the background density. Theoretically, above the reference level considered to perform the Bouguer Correction (0 m respecting to ellipsoid in this case), the background density is equal to the reference density (or reduction density), i.e. 2.67 gr/cm3 in this case. To clarify the point, we include the value in the corresponding sentence.

Section 3.2.1. Too few specifications about the method associated with forward gravity modeling. Please provide more information about the basics of the GravGrad modeling scheme of Maksymowicz et al. (2015) and its specific application to the study region, including the original geometry of each section, downward extent, background density structure, how densities of different bodies are assigned and validated, how the constraint information is incorporated into the model, are the geometries of the bodies modified interactively?

R-. The basics of the GravGrad modeling scheme is largely explained in Maksymowicz et al. (2015), but we include some modification in the section 3.2.1 to clarify the interactively model procedure. On the other hand, details on the independent information used to constraint the model is listed in the section 3.3.1. Note that in the new version we corrected the section numbers.

Section 3.2.2 3D gravity inversion. I am confused with the target of this model, mostly with the depth of the 3D model and what are you trying to obtain. In addition, with the confusing information noted in lines 171 to 176 regardinf the spatial extend of the inversion space, it is not clear what is the maximum depth of the model, it seems to be only few kilometers below Earth surface? If this is the case, can you please justify the application of this tool considering that you are interested in the crustal-to-lithospheric scale structure of the subduction zone?

R-. As is pointed in the introduction, this work aims to study the forearc density structure at regional scale by 2D forward modeling and, at local scale, by 3D density inversion of onshore data. Then, we are interested not only in the crustal-to-lithospheric scale, but also in the upper crustal density structure below the sedimentary units of central depression, where few geophysical or geological drilling campaigns has been performed.

While the regional models provide a possible solution of the density structure of the subduction zone (under the available independent constraints), the 3D local inversion can be seen as an automatic solution (independent of the forward 2D method) of the upper crust density contrast bodies (above ~20 km depth according to the results) which explain the gravity anomalies respecting to a regional linear trend, i.e. deeper anomalies (at lower crusts and mantle, for instance) are mostly contributing to regional linear trend at the scale of 3D local study. To show this approximate depth limit (~20 km), new version of supplementary material presents 3D views with the model density obtained to 50km of depth. It is important to highlight that the 3D inversion method is completely different numerical approach to 2D forward method, but the obtained density structure is similar (at common depths), which reinforce the modeled characteristics of density distribution in the upper crust.

324-325. How is defined the base of the inversion model at 20 km depth? This is not clear in section 3.2.2 3D gravity inversion, where it seems that the maximum depth is something larger than 7.5 km, but 20 km looks like too deep for this kind of model. Please clarify.

R-. The inverted space (The 3D mesh) has 67x80x102 blocks (in X, Y, Z direction), and vertically, the cell size gradually grows from 100 to 1500 m, reaching 70 km in depth (see section 3.2.2 and its modifications). In this inverted space, the 3D inversion algorithm founds anomalies to ~20 km in depth (see Figure 7). Which means that deeper anomalies are mostly contributing to regional linear trend, as was mentioned before. This idea was included in the new version of the manuscript.

349-365. How do you justify the spatial extend and the resolution of the model? Is there resolution and/or sensitivity test that allows to trust the model results??? This point is important and critical to be solved in a new version of the paper.

R- The spatial resolution of the model is restricted to space discretization mesh, which have 3000 m x 3000 m size (horizontal) and vertically gradually grows from 100 to 1500 m, in accordance with the developer's recommendation (https://www.eoas.ubc.ca/ubcgif/iag/sftwrdocs/technotes/faq.htm). On the other hand, the horizontal spacing (3000 m x 3000 m) of inverted gravity grid (and space mesh) was selected to generate a regular input data signal whit similar spacing than the average spacing of available measurements. Then, density anomalies with sizes at the order of the horizontal and vertical discretizing cannot be considered, which is the case of our interpretation where we only describe and analyze the main observed anomalies and trends, much larger than model discretizing. Numerical experiments were developed to observe the sensitivity of the solution under the variation of parameters and data, some of them are show in the new version of the supplementary material. This analysis indicated that main interpreted characteristics of the solution remains unaltered under variations of scale length parameters, and the introduction of artificial noise in the input gravity data. Beyond these numerical considerations, as in any gravity inversion, the presents not uniqueness in the solution, which is an important methodological reason to use independent model methodologies (2D forward and 3D inversion, in this case) and contrasting the results with independent available and new geophysical constrains. Finally, some modifications were included to give more details of the density inversion (section 3.2.2).

- Improve quality and clarity of interpretations and discussion.

Sections 4 and mostly 5 need some improvement in terms of the description, interpretation and discussion of results that for some cases is too confusing. The discussion would benefit from a clear separation of different points, like the geological nature of dentiy anomalies and influence of crustal structure on megathrust seismogenesis.

255-260. Please explain how the gravimetric lineaments were identified; is this just a visual exercise? You would need to justify these identifications, which seems to be quite whimsical. Is it really necessary to include these lineaments? Perhaps a good description of how the recognized (published) crustal faults and geologic lineaments correlate with gravity is a better idea in this case.

R- These lineaments were visually interpreted, which in our opinion is a valid procedure to provide a qualitative description of the gravity signal. Similar qualitative exercise is often used in bathymetric/topographic, earth magnetic field, gravity and seismic studies, and other geophysical/geological analysis to highlight linear features in the signal that could be related (or not) with hidden structures and other geological features at depth. To make easier this qualitative interpretation, it is a common practice to generate set of derivative filters of the original signal to highlight short wavelength features. A set of figures with the interpreted gravity lineaments and derivative filters is presented in the new version of the supplementary material (first derivative to the west, first de first derivative to the north, directional derivative to the northeast, slope gradient and analytical signal).

In the Figure 3b the interpreted gravity lineaments are drawing with dotted lines on the CBA grid and Figure 3a shows also the CBA grid without these interpreted features in order to facilitate the direct evaluation of our interpretation by the reader. Regarding the relation of interpreted gravity lineaments with structure published in the zone we explicitly state that "The gravity lineaments confirm the location of fault zones previously identified at the surface (SERNAGEOMIN, 2003; Melnick and Echtler, 2006), suggesting their continuity through the forearc and, in some cases, their seaward extension (e.g., Valdivia-Futrono lineament, VFL in Fig. 3b)". In fact, published fault and structures and their names are presented in Figure 3 (blue lines). The clear relation between previously identified crustal faults with some of the interpreted gravity lineaments is the primary reason to show this qualitative interpretation, because some of these lineaments could be confirmed (or not) as crustal structures in future works. In the new version of the text, we explicitly clarify the visual/qualitative type of this interpretation.

280-289. Note that the CBA high called H1 in Fig 3a does not appear in P1, but the modeled density high called H1 in Fig. 4 is present in all the profiles: it is confusing to use the same nomenclature for gravity and density anomalies in this case, perhaps it is better to change these names.

R- We agree, in the new version of the manuscript L1, L2 and H1 were maintained for gravity anomalies and D1 and D2 were used for density anomalies.

292-295. Why a high-density anomaly shall be related to the volcanic arc? I would expect the opposite since magmatic bodies and the entire plumbing system underneath volcanoes should have a much lower density than the country rock. Please explain this.

R- We agree. The phrase is confusing because we originally use the volcanic arc as a geographic reference for the roughly location of D2 (or H2 in the original text). D2 is a deeep feature in the upper crust, and then, it is not necessarily related to the plumbing system, location of magma bodies and structural of the upper portion of the crust. However, we observed that "LOFS approximately correlates with the western limit of D2 in profiles P1_Toltén, with the eastern limit of D1 in profile P2_Unión and with the eastern border of D2 at profiles P4_LLanquihue and P5_Chepu", which in fact in suggests an structural relation between the deep geometry of the high density anomalies (D1 and D2) and LOFS, which in turns, have a close relation with active volcanic systems in the region (Lara and Folguera, 2006; Sánchez et al., 2013; Díaz et al., 2020). On the other hand, the results of 3D model show that "Most of quaternary volcanoes are located in zones with negative density contrast below 5 km depth (Fig. 6). The upward migration of magmas should generate local weakening zones in the overriding plate, and consequently, the continental crust in the active volcanic zone should present pervasive fracturing, fluid migration and lower density". In the new version of the manuscript, we change this paragraph including references for the relation between LOFS and the active volcanic arc.

Are you sure Contreras-Reyes et al. (2008 and 2010) mention that the age of the paleo-acreccionary prism is Mesozoic to Tertiary? The metamorphic complexes in the region are of clear Late Paleozoic to Triassic age.

R- This paragraph refers to the middle wedge unit (MWU), seaward from outcrops of Paleozoic-Early Mesozoic accretionary complex (WS/ES). The eastern portion of MWU should be the offshore continuation of WS/ES, however the western portion of MWU could be formed by a younger "paleo-accretionary prism" (Jurassic age as is suggested by Contreras-Reyes et al., 2008) which even could include younger basal accreted material, coeval with Miocene erosional phase (Contreras-Reyes et al., 2010). On the other hand, and as is indicated in the text, there are not direct information (boreholes) to confirm the age of the continental basement in the western portion of MWU.

444-454. For the discussion about the geological nature of the H1 anomaly you should consider Plissart et al. (2019; https://doi.org/10.1016/j.lithos.2019.03.023) and references there in, which shows that the metamorphic basement associated to the WS south of 39ºS includes a great proportion of mafic and ultramafic (serpentinites) rocks that were incorporated inside a subduction channel during the Carboniferous. As H1 gravity anomaly in Fig. 3 extend southward into the Chiloe island where only Late Paleozoic metamorphic rocks has been described (i.e. no Devonian intrusive rocks related to the Chaitenia island arc), one could imagine that H1 (both gravity and density anomalies) are mostly related to the dominance of these lithologies. However, it is important to consider the evolutionary interpretation of Plissart et al. (2019) because they actually link the occurrence of these (ultra)mafic rocks to the creation of an island arc and backarc region disconnected to the main Gondwana margin during the Devonian, similar to the original idea of Herve et al. further south. This could support your interpretation, but you should complement the argumentation already exposed in this section.

R- We agree, the evolutionary interpretation of Plissart et al. (2019) is relevant for the discussion and was included in the new version of the manuscript.

456-457. I don´t see the supposed correlation between volcanoes and negative density contrast in Fig. 6; please mark clearly the volcanoes in this figure and provide actual values of density contrast to judge about it. This is

also in contradiction with what is exposed in lines 292-295, i.e. a correlation between high density and the presence of the volcanic arc and LOFZ. Please clarify.

R- As was mentioned before, D1 and D2 are crustal anomalies, not necessarily associated to shallow magmatic system below the active volcanoes, but their limits could be related to deep geometry of LOFS, which in turn have some control on the active volcanism in the region. Al lower scale, active volcanoes are located in zones with low density contrast (in general < 0.0 gr/cm3), inside de D2, as is observed in figure 6 (see red ellipses below).

[Figure]

480-481. A better reference for the Melinka earthquake in terms of describing the physical properties of the forearc is Moreno et al. (2018; https://www.nature.com/articles/s41561-018-0089-5). This discussion about the nature of seismic segments along the Valdivia earthquake segment could benefit from including findings and ideas of Molina et al. (2021).

R- Moreno et al. (2018) and Molina et al (2021) were referenced in the new version of the manuscript.

Minor points

27-28. Include Molina et al. (2021)

R- Molina et al (2021) were referenced in the new version of the manuscript.

31-. Reference is Tassara (2010). Add Molina et al. (2021)

R- Corrected in the new version

34-. Include Molina et al. (2021)

R- Molina et al (2021) were referenced in the new version of the manuscript.

50-. Replace "Fithermore" by "Furthermore"

R- Corrected

54-. Replace "at the south of…" by "southward of…"

R- Corrected

Fig. 1. Colors of geologic units are somehow masked by topography; perhaps you can either choose a color table with grey tones for topo/bathy (changing colors for metamorphic units in this case), or to use topo/bathy only in B along with the potential trace of basement domains, leaving in A the geology, structures, slip, fracture zones. By the way, the dashed outline in B is very usefulness and awkward, please remove it.

R- Fig.1 was improved according to the suggestions of reviewers.

67-. Solve "Schematic map of map of basement…"

R- Corrected

72-. Put "cm/yr" in "convergence rate (6.6 cm, Angermann, 1999)"

R- Corrected

85-. Replace "fiction" by "friction"

R- Corrected

88-. Replace "…three mayor trenches parallel morphoestructural units…" by "…three mayor trench-parallel morphoestructural units…"

R- Corrected

105-107. Rephrase this sentence, it is awkward.

R- Rephrased as: "This paired metamorphic belt is observed continuously at the CC, but the width of their outcrops varies along the margin (see Fig. 1a). Between ~38°S and 40°S, and southward of ~ 41.5°S, outcrops of WS are observed eastward, near the western limit of PC. Thus, between ~40°S to ~ 41.5°S, the eastern limit of these units is not defined due to the presence of the CD deposits and could form most of the forearc basement or it could be confined near the coast."

119-. Replace "Devonic" by "Devonian"

R- Corrected

120-. Replace "Ch in Fig. 1b" by "Ct in Fig. 1b"

R- Corrected

121-. Remove "Ch in Fig. 1b" (is already indicated in line 120).

R- Corrected

120-121. A good and updated reference for this is Rapela et al. (2021, https://doi.org/10.1016/j.gr.2021.04.004)

R-Rapela et al. (2021) was referenced in the new version of the manuscript

127-. Replace "include:" by "includes:"

R- Corrected

147-. Replace "schema" by "scheme"

R- Corrected

149-. Replace "…it should be modelling considering…" by "…it should be modelled considering…"

R- Corrected

165-. Replace "3.2.1 3D gravity inversion" by "3.2.2 3D gravity inversion"

R- Corrected

172-. Replace "…blocks (in X, Y, Z direction), respectively." by "…blocks (in X, Y, Z direction, respectively)."

R- Corrected

195-. Replace "3.2.1 Available geophysical information" by "3.3.1 Available geophysical information"

R- Corrected

210-. Replace "3.2.1 Electromagnetic methods to constrain gravity measurements" by "3.3.2 Electromagnetic methods to constrain gravity measurements"

R- Corrected

Figure 3. Replace "Grvimetric Lineaments" by "Gravimetric Lineaments" in the legend of the figure.

R- Corrected

Figure 4. This is a bit confusing, and you could consider some of the following suggestions: 1) Put each gravity profile with the corresponding density profile, so one can appreciate the correlation between anomalies and the modeled density structure. 2) Try to separate or identify the original CBA from the modeled anomaly, since in the current Fig. 4a is impossible to recognize it. 3) Use a different name for H1 and H2 because it is confused with H1 of Fig. 3a although they are not the same. Fig 4d has a problem with numbers in the x-axis, please correct it.

R- This figure was designed to allow a direct comparison of all density models in a single image. If we add individual panels with gravity data above profiles it is necessary to greatly reduce the scale of each panel, which in our opinion, is not convenient for model comparison. Then, we prefer to maintain the design of the figure (correcting the errors), but also including individual figures with corresponding gravity panels in the new version of the supplementary material.

281-282. Replace "… and increase to deep" by "…and increase downwards"

R- Corrected

291 and elsewhere. Please do not use "before" and/or "after" to refer to east-west locations with respect to a given feature, better use westward or eastward!

R- Corrected in the new version of the manuscript.

292-. Replace "important toconsider" by "important to consider"

R- Corrected

Figure 5. Please include the original CBA and the regional field obtained as a polynomial representation that is extracted from the observed CBA in order to get the residual CBA.

R- Fig. 5 is highlighting the input data of 3D inversion (RBA) and the final fitting between observed and modeled data. As the regional trend (not inverted) is only a single plane, we include its equation and a figure with CBA, RBA and linear regional trend in the supplementary material.

349-. What is the DC? Do you mean CD I guess.

R- Corrected

385-. Replace "see and interpretative schema at Fig. 8a" by "see an interpretative scheme at Fig. 8a"

R- Corrected

393-. Replace "bangs et al., (2020)" by "Bangs et al. (2020)".

R- Corrected

429-. Is there any specific references for this supposed west-dipping reverse fault that puts CC in tectonic contact with CD??

R- This sentence was removed in the new version. Few lines above (and also in section 4.1) we include the references:

Melnick, D. and Echtler, H. P.: Morphotectonic and geologic digital map compilations of the south-central Andes (36–42°S), In: Oncken, O., Chong, G., Franz, G., Giese, P.,Götze, H.-J., Ramos, V.A., Strecker, M., Wigger, P. (Eds.), The Andes – Active Subduction Orogeny. Frontiers in Earth Science Series, Vol. 1. Springer-Verlag, Berlin, Heidelberg, New York, pp. 565–568, 2006.

Hackney, R., Echtler, H., Franz, G., Götze, H. J., Lucassen, F., Marchenko, D., Melnick, D., Meyer, U., Schmidt, S., Tašárová, Z., Tassara, A., and Wienecke, S.: The Segmented Overriding Plate and Coupling at the South-Central Chilean Margin (36-42°S), In:Oncken, O., et al. (Ed.), The Andes-Active Subduction Orogeny, Frontiers in Earth Sciences. Springer-Verlag, Berlin, Heidelberg, New York, pp. 355–374, 2006.

Encinas, A., Sagripanti, L., Rodríguez, M.P., Orts, D., Anavalón, A., Giroux, P., Otero, J., Echaurren, A., Zambrano, P. and Valencia, V.: Tectonosedimentary evolution of the Coastal Cordillera and Central Depression of south-Central Chile (36°30′-42°S), Earth-Science Reviews,Volume 213,103465, ISSN 0012-8252, https://doi.org/10.1016/j.earscirev.2020.103465. 2021.

430-. Remove one of both "depth" in the sentence "the depth contact between CC domain and H1 at depth"

R- Corrected

449-. Please close the parenthesis after "(Hervé et al., 2016; 2018"

R- Corrected

466-. Replace "intreseismic deformation" by "interseismic deformation"

R- Corrected

468-. Please provide relevant reference for this sentence.

R- We referenced to Scholz, 1998; Perfettini, H., and Avouac, 2004; Tassara, 2010; Moreno et al., 2018 and Im et al., 2020 in this paragraph.

469-. Replace "fractured and or metamorphic" by "fractured and/or metamorphic"

R- Corrected

474-. Replace "…should modified the…" by "…should modify the…"

R- Corrected

490-. This is also observed by Molina et al. (2021) and you can used to reinforce this idea. In this line please replace "This siggests oversaturate fluid…" by "This suggests that over-saturated fluid…"

R- Corrected.

494-. Replace "Several authors have siggested" by "Several authors have suggested"

R- Corrected

---

## Author Comment (AC4)

**Response to comments by Dr. Carla Braitenberg on the manuscript "Forearc density structure of the overriding plate in the northern area of the giant 1960 Valdivia earthquake" (se-2021-53)**

We thank the detailed revision on our work from Dr. Carla Braitenberg (https://doi.org/10.5194/se-2021-53-RC3). In this document we write responses in blue after the reviewer's comments (black). Numerous writing corrections annotated by the reviewer in an attached pdf file were included in the new version of the manuscript (highlighted in green).

Sincerely

The authors

The research aims at defining a density structure of the continental forearc in the northern segment of the 1960 Valdivia earthquake, to date the highest magnitude event ever recorded with seismologic and geodetic instruments. The area of study is of general interest due to this landmark event, which generated free oscillations of the earth observed for the first time. The authors present a density model that aims to explain the observed terrestrial and satellite gravity data and to demonstrate a segmentation of the continental wedge, both along and across the subduction margin. The authors propose that the inhomogeneous structure of the overriding plate controls the process of stress loading during the interseismic period, due to rigidity variations. This point is interesting, but at the present stage of the work the link between the density structure and the rigidity variations and stress pattern is suggested qualitatively, but lacking a quantitative estimate. The general assumption made in the

Discussion chapter is, that denser crust is more rigid. This assumption could be supported by some numbers, so as to define what the expected changes in elastic parameters are and which the uncertainties. In the discussion it is mentioned that in the northern profile, P1_Toltén, the density model can be compared with the velocity model reporting Vp values- the authors could use density and velocity along the profile to calculate the elastic parameter changes, in order to support their hypothesis.

R.- We interpret that the general increase of density from MWU (and CC domains) to the east can be related to an increase of rigidity and/or a decrease of fracturing. This interpretation is supported by the correlation between high density anomaly increase of Vp, decrease of Vp/Vs ratio (Dzierma et al., 2012a), and also, high electrical resistivity of MT studies (Kapinos et al., 2016). Inside the region highlighted by white contour in Fig.4b (P1_Toltén profile), Dzierma et al. (2012a) shows Vp/Vs values lower than 1.74, contrasting with values higher than 1.78 eastward and westward. In same region Vp and Vs models reach values ~4% and ~8% higher than surrounded regions, respectively. Then, at least at the profile P1_Toltén, the correlation between D1 high density anomaly and changes in the elastic properties is clearly observed. Considering this Vs velocity anomaly and moderate increase of density of associated to D1 anomaly of about 0.05 gr/cm3 (Fig4a and 7), we estimated an increase of shear modulus asociated to D1 at the order of 20% in comparison to surrounded regions (at the same depth). These values were included in the new version of the manuscript. Unfortunately, the calculation of a grid of shear modulus based on our density grid and the velocity values is not a direct procedure because Dzierma et al., 2012a did not include the velocity grid values in tables (only colored figures are available). However, it is and interesting suggestion, and should be included in future works in zones where Vs models of forearc are available.

The authors further propose that the varying width along the margin of the MWU and CC domains could be due to varying friction at the interplate boundary. Since friction at a sliding interface is the product of the normal stress component to the surface and the friction coefficient, I wonder if the authors could use the density model to calculate the normal stress and then make implications on the frictional coefficient and possible presence of fluids.

R.- We propose that "the widening of MWU and CC domains to the north of ~42ºS could favoured high friction in the deep region of interplate boundary (bellow CC domain) and a relatively low friction in the seaward

portion of MWU", due to a possible dewatering of the deep region associated to deformation style of basal accretionary complexes in MWU and CC domains (Menant et al., 2019). Then, we did not interpret that the deformation style (or width) of MWU and CC domains is due to friction variation at the interplate boundary.

As in our previous works in the Chilean margin (northward from the study zone, Maksymowicz et al, 2015; 2018) it is possible to calculate the total vertical load over the interplate boundary, as an estimation of normal stress component. However, the frictional coefficient (effective basal friction coefficient, Dahlen, 1984) is estimated from the morphology of the continental wedge but it is virtually independent of the continental wedge density (Maksymowicz, 2015). On the other hand, as the shear stress along the interplate boundary cannot be determined independently, the friction coefficient cannot be derived from estimations of vertical load over the contact. This makes impossible (or highly speculative) to link mathematically the estimations of effective basal friction coefficient with the estimations of vertical load. However, these independent models (density models and models of effective basal friction coefficient) are suggesting changes in the friction variations and fluid migration along the megathrust.

Concluding, I propose the authors use their density model to quantitatively support their implications on the seismotectonics. In the following some specific problems are addressed.

Specific problems

It is discussed that gravimetric lineaments have been defined, partly based on previous publications, partly defined in the present paper. At first sight these lineaments are not seen the BGA, so the authors need to define how the determine the presence of a lineament.

R- These lineaments were visually interpreted, which in our opinion is a valid procedure to provide a qualitative description of the gravity signal. Similar qualitative exercise is often used in bathymetric/topographic, earth magnetic field, gravity and seismic studies, and other geophysical/geological analysis to highlight linear features in the signal that could be related (or not) with hidden structures and other geological features at depth. To make easier this qualitative interpretation, it is a common practice to generate set of derivative filters of the original signal to highlight short wavelength features. A set of figures with the interpreted gravity lineaments and derivative filters (first derivative to the west, first de first derivative to the north, directional derivative to the northeast, slope gradient and analytical signal) is presented in the new version of the supplementary material.

In Figure 3b the interpreted gravity lineaments are drawing with dotted lines on the CBA grid and Figure 3a shows also the CBA grid without these interpreted features in order to facilitate the direct evaluation of our interpretation by the reader. Regarding the relation of interpreted gravity lineaments with structure published in the zone we explicitly state that "The gravity lineaments confirm the location of fault zones previously identified at the surface (SERNAGEOMIN, 2003; Melnick and Echtler, 2006), suggesting their continuity through the forearc and, in some cases, their seaward extension (e.g., Valdivia-Futrono lineament, VFL in Fig. 3b)". In fact, published fault and structures and their names are presented in Figure 3 (blue lines). The clear relation between previously identified crustal faults with some of the interpreted gravity lineaments is a primary reason to show this qualitative interpretation, because some of these lineaments could be confirmed (or not) as crustal structures in future works. In the new version of the text, we explicitly clarify the visual/qualitative type of this interpretation.

The English Grammar must be improved- I attach a pdf with many small corrections

R.- We appreciate the detailed revision. Numerous writing corrections annotated by the reviewer in an attached pdf file were included in the new version of the manuscript (highlighted in green).

The relation between the gravity field and the mega-faults for the Andean Subduction margin has been discussed before, but these papers are missing- I propose these findings shall be included in the introduction. They have been produced by Researchers as Orlando Alvarez and co-authors (https://scholar.google.com/citations?user=MDsDjEcAAAAJ&hl=en). These have also demonstrated the interaction of the morphology of the subducting topography on the angle of subduction.

R.- Álvarez et al., 2014 is cited in the new version of the manuscript.

Line 92: give a few words on how the right lateral strike slip system relates to the oblique subduction. Give some more details on the crustal seismic fault mechanisms present in the area.

R.- Added in the new version of the manuscript.

Line 139:" normal gravity correction (subtracting the theoretical gravity of the WGS-84 ellipsoid), Free-Air, Bouguer, and Terrain corrections": give more details, as different standards exist. Give the formulas for the corrections. GPS give elliposidal heights, which geoid was used to obtain normal heights? Did you define gravity anomlay or disturbance? up to which radius did you make the topographic correction? Did you first calculate simple Bouguer and then terrain correction? did you use a higher resolution DTM for the near field? What is the estimated error on the final gravity acquisition?

R.- The paragraph was modified to include details of the process of gravity data reduction and estimated data error.

We applied the process broadly used in geophysical studies to derive the Complete Bouguer Anomaly from direct surface gravimetric measurements (Blakely, 1995; Lowrie, 2007). It is clear that in geodesy studies there are different definitions of gravity anomalies and corrections (Hackney and Featherstone, 2003). According to these authors, the subtraction of normal gravity of the ellipsoid (calculated in the surface of the ellipsoid) from the absolute gravity registered in the surface of the earth is defined as scalar disturbance, and is one of the firs step applied in the process of gravity data reduction.

R. I. Hackney, W. E. Featherstone, Geodetic versus geophysical perspectives of the 'gravity anomaly', *Geophysical Journal International*, Volume 154, Issue 1, July 2003, Pages 35–43, https://doi.org/10.1046/j.1365-246X.2003.01941.x

Did you have coincident old and new datapoints and what is the mean difference and standard deviation? What is the standard deviation compared to the GOCE field? Notice you must low-pass filter your data to make the differences with GOCE.

R.- As is mentioned in the text, the new gravimetric data were distributed to fill in some observed gaps in onshore studies, and to complement and validate gravity and topographic information from old stations. The new gravity measurements were tied to the absolute gravity stations available in the study zone (International Gravimetric Bureau (BGI), https://bgi.obs-mip.fr/). The average difference in absolute gravity values between new stations and every old station located closer than 1 km is 0.29 mGal, reflecting that old data are consistent with new data, even considering changes of elevations between not coincident points in rough topography areas.

We include satellite derived data in the merged gravity database. In fact, the last version satellite derived data of Sandwell and Smith (https://topex.ucsd.edu/cgi-bin/get_data.cg) is used to cover marine gaps and regions to the south of 42ºS. In a previous work, Maksymowicz et al. (2015) confirmed that the shape and amplitude of this satellite data (i.e., all relevant features for the modeling scale) are preserved in comparison with the Free-air data obtained by the direct acquisition of marine gravimetric lines. This favore the use of this satellite grid to be merged with marine and onshore surface gravity measurement. As is pointed by the reviewer, GOCE provide a low-frequency gravity anomaly, providing less information of density structure of the subsoil, which in turns is the primary objective of this work.

Line 170: "This linear regional gravity trend is mostly related to a deep continental root below the Andes, eastward from inverted gravity" Linear trend of gravity field: do you mean crustal root and subducting lithospheric plate? You could at least calculate an isostatic root and calculate its gravity field to show that the linear plane resembles the isostatic field. The effect of the subducting plate has been calculated before and could be used or at least mentioned to estimate its effect on your field.

R.- This sentence was removed in the new version of the manuscript. While 2D models provide a possible solution of the density structure of the subduction zone (under the available independent constraints), the 3D local inversion can be seen as an automatic solution (independent of the forward 2D method). As is pointed in the new version of the text: "The 3D inversion modelled the input Residual Bouguer Anomaly (Fig. 5a) with high precision, as is observed in Fig. 5b, where differences between modelled and observed data are, in general lower than ± 1 mGal. The results show density contrast anomalies to about 20 km depth (Fig. 6, 7 and supplementary material), which means that deeper anomalies are mostly contributing to regional linear trend of the CBA at the scale of 3D local inversion". It is important to highlight that the 3D inversion method is completely different in numerical approach to 2D forward method, but the obtained density structure is similar (at common depths), which reinforce the modeled characteristics of density distribution in the upper crust and the considered linear regional trend.

Line 184: define density background model

R.-. The modeling of residual gravity provides contrasts of density respecting to the background density. Theoretically, above the reference level considered to perform the Bouguer Correction (0 m respecting to ellipsoid in this case), the background density is equal to the reference density (or reduction density), i.e. 2.67 gr/cm3 in this case. To clarify the point, we include the value in the corresponding sentence.

Line 195: here you mention the geophysical models for slabs and crustal thickness- it is not clear why previously you claim to subtract a plane that represents crustal roots of the Andes- the longest period field should be explainable by the slab and crustal thickness variation. Could you use these models to correct for the longest gravity field wavelengths?

R.- As was pointed before, the independent information of slab structure and geometry was used as a constraint for 2D density models. With a different modeling approach, 3D inversion considers the previous subtraction of a linear regional trend to generate a Residual Bouguer Anomaly. Then, all gravity signals, different from the linear trend, generated in the slab, lower continental crust and/or upper mantle are included in this Residual Bouguer Anomaly. Accordingly, as the results of the inversion show density contrast anomalies to about 20 km depth, and a mostly homogeneous layer below (see supplementary material), the deeper anomalies are mostly contributing to regional linear trend of the CBA at the scale of 3D local inversion. On the other hand, the 3D inversion present similar density structure in comparison with 2D model (at common depths) which means that the gravity signal of the deeper anomalies modeled in 2D profile (whit the original Complete Bouguer Anomaly) are well approximated by the linear regional trend.

In general, use SI units, that is kg/m^3

 R.- We change the density unit to standard gr/cm3

---

## Author Response (AR2)

**Response to the editor review on the manuscript "Forearc density structure of the overriding plate in the northern area of the giant 1960 Valdivia earthquake" (se-2021-53)**

Dear editor,

We would like to submit a new version of the manuscript se-2021-53. The new version of the paper includes minor changes (highlighted in green) to fix some writing errors and to include the suggestions from the reviewer Dr. Carla Braitenberg. Below, we write responses in blue after the reviewer's comments (black).

Best regards

The authors

Dear authors, I appreciate you have made considerable effort to improve the manuscript following the suggestions of the reviewers. Now that the data processing of the new gravity data has been explained, I find there is a problem which should be either corrected or made clearer, and justified, ideally also estimating the error which is introduced by using ellipsoidal heights and not sea level heights in the free air height correction. A further issue is the shift of different databases, which is an important point and should be documented giving the shifts in a table and some statistics. A few detailed comments follow:

Morphoestructural-> morpho-structural

R.- Morphoestructural was replaced by Morpho-structural

L. 152: Free-air correction of all onshore data was

calculated as 0.3086h (mGal), where h is ellipsoidal high in meters (Lowrie, 2007). The terrain correction of all data was calculated following a combination of the algorithms proposed by Kane (1962) and Nagy (1966) and with high resolution SRTM elevation grid.

h is ellipsoidal high in meters-> h is ellipsoidal height in meters

R.- This error was corrected through the entire text

☐ Please check- have you really used ellipsoidal heights for height correction for new data? This would correspond to calculation of gravity disturbances and not anomalies-

R.- We use ellipsoidal heights for all data processing. In particular, new differential GPS data were acquired and processed by us to obtain ellipsoidal heights.

th old data probably are anomalies, e.g. they used heights above sea level for the correction. Further you use SRTM for the topography correction, but these heights are given above sea level. Please check this point. You can fnd a recent discussion on the corrections, and use of normal or ellipsoidal heights in this paper published in:

☐ The first pan-Alpine surface-gravity database, a modern compilation that crosses frontiers Pavol Zahorec, Juraj Papčo, Roman Pašteka, Miroslav Bielik, Sylvain Bonvalot, Carla Braitenberg, Jörg Ebbing, Gerald Gabriel, Andrej Gosar, Adam Grand, Hans-Jürgen Götze, György Hetényi, Nils Holzrichter, Edi Kissling, Urs Marti, Bruno Meurers, Jan Mrlina, Ema Nogová, Alberto Pastorutti, Corinne Salaun, Matteo Scarponi, Josef Sebera, Lucia Seoane, Peter Skiba, Eszter Szűcs, and Matej Varga. Earth Syst. Sci. Data, 13, 2165–2209, https://doi.org/10.5194/essd-13-2165-2021, 2021.

R.- Old database (published, merged and described by Schmidt and Götze,2006) merged data from different campaigns and years. Oldest campaigns probably use "sea level" heights, but more recent data included dGPS measurements. Due to this inhomogeneity, we compile a "Bathymetric/topographic database merges onshore elevation grid (SRTM elevation grid, Jarvis et al., 2008) and swath bathymetry data of the studied zone (Flueh and Grevemeyer, 2005), complemented by Global Topography V18.1 (Smith and Sandwell, 1997)" as is

indicated in text. Then, in the case of the old data, Free-air and Bouguer correction was performed with this ellipsoidal height database. This point was clarified in the new version of the manuscript.

☐ Please give the maximum radius you used for the topographic masses in Bouguer correction.

R.- As is pointed in the new version "The terrain correction includes topographic data located up to ~300 km around each station".

L. 157: The spatial coverages of different gravity databases (satellite, marine, and onshore) present areas of interception (Fig. 2) where they can be compared to determine the average gravity differences (constant average shifts).

☐ Please give some statistics about the shifts you introduce into the data. Justify why you think the terrestrial data are at the correct value, the marine data not.

R.- In the new version of the manuscript we include the shifts applied to merge gravity data. We did not think that the marine data are incorrect in comparison to the terrestrial data. However, we prefer to tide all gravity observation to our onshore observations because there we have direct gravity measurements linked to the absolute gravity stations. On the other hand, in terms of density modeling, the selection of this "common gravity level" is not relevant because finally represent a constant value in the Complete Bouguer signal which theoretically does not affect the result of the model.

L. 159: he Free-air -> The

R.- corrected in the new version.